# Wide field-of-hearing metalens for aberration-free sound capture

Dongwoo Lee [1,6], Beomseok Oh [1,6], Jeonghoon Park [1], Seong-Won Moon[1], Kilsoo Shin[1], Sea-Moon Kim [2] & Junsuk Rho [1,3,4,5] ✉

Metalenses are instruments that manipulate waves and have exhibited remarkable capabilities to date. However, an important hurdle arises due to the severe hampering of the angular response originating from coma and field curvature aberrations, which result in a loss of focusing ability. Herein, we provide a blueprint by introducing the notion of a wide field-of-hearing (FOH) metalens, designed particularly for capturing and focusing sound with decreased aberrations. Employing an aberration-free planar-thin metalens that leverages perfect acoustic symmetry conversion, we experimentally realize a robust wide FOH capability of approximately 140° in angular range. Moreover, our metalens features a relatively short focal length, enabling compact implementation by reducing the aperture-to-hearing plane distance. This is beneficial for space-efficient source-tracking sound sensing. Our strategy can be used across various platforms, potentially including energy harvesting, monitoring, imaging, and communication in auditory, ultrasonic, and submerged environments.

Sound waves are omnipresent in our daily lives, and harnessing their energy in a specific domain holds great significance for sound-focusing or sensing applications, such as wireless communication, medical diagnostics, nondestructive evaluation, energy harvesting, tweezers, and imaging[1–4]. In this context, an acoustic lens plays a crucial role in achieving desired wave manipulations. For instance, an acoustically transparent curvilinear lens can be applied in ultrasound probes for medical sonography. To promote space-saving with thinner devices and low-cost manufacturing, an acoustic nonplanar[5,6] or planar[7,8] Fresnel lens has been introduced with concentric grooves or annular slits to improve the focusing capabilities. Interestingly, employing the curved-space framework in elastic domains makes exceptional wave-bending capabilities by refractive index-curvature equivalence, such as those exhibited by wormholes, black holes, and retroreflectors, more feasible[9–11].

Metamaterials, made of subwavelength building blocks, have demonstrated considerable potential in achieving extraordinary focusing capabilities. In particular, the concepts of a superlens and hyperlens counterintuitively go beyond the Abbe diffraction limit, which was realized in both optical[12–15] and acoustic regime[16–20]. Both lenses can magnify sub-diffraction spatial information contained in evanescent waves and carry them to the imaging plane of interest, enabling super-resolution imaging by either a negative refractive index or a strong anisotropic medium.

On the other hand, metalenses may offer great opportunities as a planar, thin, cost-effective, and highly efficient imaging platform[21–29]. This planar compact form factor enables precise wavefront-shaping, where each artificial element within the device is carefully designed in a space-parameterized manner. Notably, optical metalenses have recently attracted significant attention for enabling wide field-of-view

[1]Department of Mechanical Engineering, Pohang University of Science and Technology (POSTECH), Pohang 37673, Republic of Korea. [2]Ocean and Maritime Digital Technology Research Division, Korea Research Institute of Ships & Ocean Engineering (KRISO), Daejeon 34103, Republic of Korea. [3]Department of Chemical Engineering, Pohang University of Science and Technology (POSTECH), Pohang 37673, Republic of Korea. [4]Department of Electrical Engineering, Pohang University of Science and Technology (POSTECH), Pohang 37673, Republic of Korea. [5]POSCO-POSTECH-RIST Convergence Research Center for Flat Optics and Metaphotonics, Pohang 37673, Republic of Korea. [6]These authors contributed equally: Dongwoo Lee, Beomseok Oh. ✉e-mail: jsrho@postech.ac.kr

(FOV) structured light[30–32]. Ongoing efforts are being made to further enhance their capabilities for applications such as holographic displays[33] and light detection and ranging (LiDAR)[34]. These endeavors continue to be actively pursued to address practical applications in various fields. Generalizing the concept of a wide FOV to the acoustic regime, if realized, holds promise with potential applications in remote sound collection in air and sound navigation and ranging (SONAR) sensing in underwater environments. Substantial progress in the acoustic counterpart has shown that considerable focusing capabilities can be achieved[35–52]. However, the concept of a wide field-of-hearing (FOH), analogous to a wide FOV, has not been introduced in the field of planar acoustic metalenses, and corresponding devices demonstrating this capability have not been realized thus far (Supplementary Table 2). Existing acoustic metalenses ever faced a fundamental limitation, specifically, the restricted coverage of the FOH and degraded resolution caused by angle-induced third-order (or Seidel) aberrations (e.g., coma, field curvature, and astigmatism). These aberrations typically hinder the focusing efficiency with an increase in the angle of incidence (AOI). Here, we propose a new class of acoustic metalens by introducing the concept of a wide FOH (WFOH) that can be used for wide-angle acoustic source-tracking platforms. The proposed design shows a broad angular response, with an AOI almost up to 70°, which is achieved using only a subwavelength single-layer metasurface comprising Helmholtz resonators (HRs) within a straight waveguide, and zigzag channels. We experimentally demonstrate the WFOH capability with highly sound-transparent and judicious phase-modulated metalens. By employing the perfect acoustic symmetry conversion, we realize an off-axis aberration-free compact sound sensing system with a reduced aperture-to-hearing distance and focal position area. Our findings have significant implications for various advanced applications, such as wide-angle acoustic sensing with high sensitivity, particle manipulation, as well as medical treatments involving high-intensity focused ultrasound (HIFU) and imaging. The successful implementation of WFOH leads to further exploring these applications and achieving improved performance.

## Results

### Problem statement and underlying physics of wide field-of-hearing

Most subwavelength planar focusing devices, known as metalenses, have been dedicated to the angle-independent (hyperbolic) and angle-dependent (ideal) form of the phase distribution. Each canonical phase profile that has been widely used in conventional metalenses is as follows: $\phi_{hyper}(r) = -k_0(\sqrt{y^2+f^2} - f)$ and $\phi_{ideal}(r, \theta_i) = -k_0(y\sin\theta_i + \sqrt{(y-y_0(\theta_i))^2+f^2} - \sqrt{y(\theta_i)^2+f^2})$, where $r$ represents the radial distance from the center of the lens, $k_0( = 2\pi/\lambda)$ is the wavenumber in free space with the wavelength $\lambda$, and $f$ is the focal length associated with the desired focal-point positions such as $y_0(\theta_i) = f\tan\theta_i$ or $f\theta_i$. In principle, space-parameterized meta-atoms arranged in a planar fashion are encoded by corresponding phase delays, enabling them to function efficiently similar to bulk and curved commercial lenses. However, achieving an ideal phase profile heavily relies on nonsymmetric behavior with respect to AOIs under oblique illumination scenarios[53,54]. This indicates the necessity for angle-dispersive properties that cannot be achieved with passive structures (see Supplementary Note 1 for details). Fully realizing an angle-induced aberration-free planar metasurface is challenging owing to off-axis aberrations such as coma and field curvature, which can degrade the focusing capability in a single-piece design with the corresponding phase library. Consequently, there is a strong desire to pursue an undistorted point spread function (PSF) without off-axis aberrations on the focal plane. The long-lasting objective is to achieve wide FOV while maintaining high efficiency within a highly compact design. To this end,

recent studies in optical fields have demonstrated that rotational symmetry can be perfectly transformed into translational symmetry to obtain high-quality PSF regardless of the AOIs[55–61]. It would be of interest to generalize the concept of wide-angle capability to acoustic waves. Envisioning the acoustic counterpart of wide FOV, we apply a similar strategy to develop the WFOH system, which can be used for a source-tracking sound sensing platform. The WFOH system exploits the quadratic form of the one-dimensional phase profile, which is expressed as follows:

$$\phi(y) = -\frac{k_0}{2f}y^2, \tag{1}$$

where $y$ represents the transverse position across the lens. The primary concept of Eq. (1) involves the transformation of a spherically symmetric gradient-index (Luneburg) lens into a flat-configuration where both the lens's geometrical shape and the focal plane can be flattened–achieving a perfect conversion from rotational symmetry to translational symmetry (Fig. 1a). If incident sound waves are projected in the $xy$-plane at an arbitrary AOI, the effective phase shift $\phi_{PS} = -k_0 y\sin\theta_i$ is incorporated, yielding the following phase distribution after transmitting through the metalens or at the exit pupil (EP) of the metalens:

$$\phi_{EP}(y, \theta_i) = -\frac{k_0}{2f}y^2 + \phi_{PS} = -\frac{k_0}{2f}(y + f\sin\theta_i)^2 + \frac{fk_0\sin^2\theta_i}{2}. \tag{2}$$

The effect of oblique incidence is evident in the transverse translation of the focal position by $f\sin\theta_i$ in Eq. (2). In the case of spherically symmetric lenses, the focal plane takes a non-planar form, requiring complex-receiving components; otherwise, it leads to field curvature aberration based on the planar focal plane. However, through a perfected symmetry conversion process, WFOH metalenses possess a planar receiving plane with predictable focal positions, enabling straightforward wide-angle acoustic sensing (Fig. 1a). Details on the derivation of the quadratic phase profile are provided in Methods. We emphasize that Eq. (2) does not imply the necessity for meta-atoms to possess angular dispersion, i.e., angle-dependent phase delay and transmission. Rather, it denotes the effective transformation from the self-contained phase shift under oblique illumination scenarios (see Supplementary Note 2 for a detailed discussion). Moreover, these characteristics enable successful realization comparable to a Fourier lens, applicable to diverse uses such as spatial filtering and compressed sensing[62], all within a compact planar form factor.

To verify the WFOH characteristics, we quantify the modulation transfer function (MTF)[63], which is a valuable metric for evaluating the imaging-resolution capability of a lens (Fig. 1b). In its one-dimensional form, MTF is obtained as the amplitude of the Fourier transform of the PSF (intensity distribution), denoted as $I$. Its expression can be written as MTF $= |\frac{\int I(y)e^{-j2\pi f_y y}dy}{\int I(y)dy}| \times 100\%$ where $I = |p|^2/Z_0$ with the pressure field $p$ and characteristic impedance of the background medium $Z_0 = \rho_0 c_0$, and $f_y$ is the spatial frequency along the $y$-axis. $f_y$ is normalized to the cutoff frequency ($2 \times NA/\lambda$) of the sound wave in the background, where the numerical aperture (NA) is defined by $\sin[\arctan(D/(2f))]$. MTF typically decreases for higher spatial frequencies, based on $2 \times NA/\lambda$ (diffraction limit), indicating that the NA imposes limitations on spatial resolution. Specifically, high $f_y$ corresponds to finer details within an object, such as edge information. Conversely, low $f_y$ relates to the broader contours and general features of the object. Therefore, the analysis of MTF and PSF is crucial for characterizing a lens's performance, particularly across various incident angles. We examine an ideal WFOH system based on Eq. (1) with a short focal length of $f = 2\lambda$, an aperture size of $D = 8.4\lambda$, and an NA of 0.9. Notably, it provides

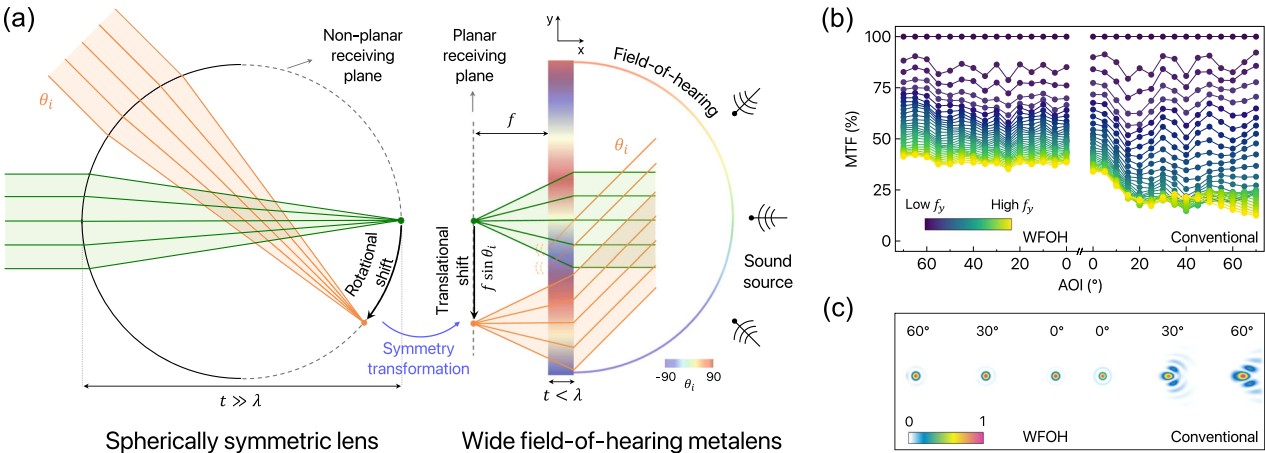

**Fig. 1 | Principle of a wide field-of-hearing system. a** Schematic illustration of the acoustic symmetry conversion that transforms rotational symmetry under oblique incidence into transitional symmetry on the receiving plane with the transverse focal shift. In this process, the spherically symmetric lens shape with a non-planar receiving plane is effectively transformed into the thin-flat lens with a planar receiving plane. FOH refers to the hearable angular range, analogous to FOV in the vision system, determined by the reception angle limit, $\pm\theta_{\max}$. **b** MTF comparison between WFOH and conventional lenses with an NA of 0.9 as a function of AOI. Due to substantial off-axis aberrations, the MTF of the conventional case drastically decreases with increasing AOIs, leading to a loss in focusing ability. Conversely, the WFOH case demonstrates consistent and flattened MTFs regardless of AOIs and spatial frequency components, indicating its wide-angle focusing capability. **c** Theoretical prediction of the focusing field for ideal 2D lenses: WFOH metalenses with small aberrations (left) and conventional metalenses with large aberrations (right) under varying AOIs.

flat-like MTF curves across the AOI with respect to a broad range of $f_y$ (Fig. 1b). This indicates that PSFs can be readily resolved for both small and large features, thereby maintaining imaging and sensing capabilities for a wide angular range, mainly due to their aberration-free characteristics. On the other hand, the conventional (hyperbolic) phase profile results in a narrow AOI working range of approximately 10°. Additionally, the MTF drops rapidly, especially for higher values of $f_y$, thus impeding regular PSF formation. In Fig. 1c, we can observe that the PSFs of conventional metalenses exhibit large aberrations at large AOIs, accompanied by low focusing efficiency and unpredictable focal positions. In contrast, WFOH metalenses have robust PSFs regardless of AOIs which indicate wide-angle focusing capability.

Without loss of generality, we now target a high NA of approximately 0.9 with fixed values of $f$ and $D$. The use of such a high NA has two benefits: improved energy gathering tied to the capacity for radiant energy, and a shorter focal length (aperture-to-hearing plane distance), leading to a more compact and robust sound reception system. In particular, the choice of $D = 8.4\lambda$ is determined by the design rule that $D$ should be larger than the effective aperture area $D_{\mathrm{eff}} = 2f$ bounded by the range $[-f+f\sin\theta_i, f+f\sin\theta_i]$, which originates from the generalized Snell's law and quadratic phase profile (see details in Supplementary Note 2). Any region beyond this range—evanescent zone—undergoes a wave-decaying process. The wavevector of transmitted acoustic waves decomposes into the wavevector components along the acoustic axis $k_x = \sqrt{k_0^2 - k_y^2}$ and $k_y = d\phi/dy$ along the transverse direction induced by the phase gradient. When $|k_y| > k_0$, $k_x$ becomes imaginary, thereby converting the sound waves into evanescent waves through total internal reflection, which does not contribute to the focusing. Owing to the translational symmetry resulting from different AOIs, the evanescent zone also undergoes a translational shift of $f\sin\theta_i$. Therefore, in determining $D = 8.4\lambda$ to span the entire range of interest $[-2f, 2f]$ while accommodating the sign of incident angles $\pm\theta_i$, it is crucial to incorporate the translational shift within the overall aperture size. This ensures that $D_{\mathrm{eff}}$ remains appropriately positioned within the specified range despite arbitrary AOIs. Interestingly, the underlying physics of the evanescent zone involves the formation of a self-adjustable aperture stop. This feature effectively suppresses additional marginal rays that might otherwise contribute to the onset of coma aberrations.

## Design principle and characterization

We employ HRs within a straight waveguide and a zigzag channel to realize the WFOH metalens, as illustrated in Fig. 2a. Since the widely used zigzag channel cannot easily achieve arbitrary transmission $T$ and phase $\phi$ due to the balance between the efficiency and mode change (see Supplementary Note 3), we integrate additional HRs for impedance matching and provide phase compensation. Each geometric configuration is utilized to achieve local resonance and Fabry-Pérot resonance in the HRs and zigzag channel, respectively, and the opened straight waveguide leverages an efficient transfer of acoustic energy with phase delays. By combining these two components into a single unit cell, we can obtain the simultaneous modulation of $T$ and $\phi$[64–66] by adjusting geometric parameters $w_4$ and $h_1$; excellent agreement between the theoretical predictions and FEM results are achieved (Fig. 2b–e). We note that our meta-atoms can also find utility in applications such as acoustic holography and particle manipulation, which require independent control of $T$ and $\phi$. However, for the purpose of the present study, we exclusively focus on phase modulation to achieve exceptionally efficient focusing over a wide range of incident angles. The theoretical description details on the meta-atom are provided in Supplementary Note 4. The geometric details are as follows: $w_0 = 0.75$, $\lambda = 75$ mm, $h_0 = \lambda/12.5 = 8$ mm, $h_L = 45$ mm, $w_1 = w_2 = h_2 = 1$ mm, and $w_3 = 10$ mm. We arrange the discretized meta-atoms along the $y$-axis, and the high sampling rate in the spatial domain ensures an aberration-free performance while preventing spatial aliasing based on the Nyquist-Shannon sampling theorem[67], where NA $\leq \lambda/(2p)$ with spatial periodicity $p$. Our meta-atom features a high resolution of periodicity (i.e., $p = \lambda/12.5$). Such a deep-subwavelength-scaled size is sufficient to cover and respond to the rapid phase gradient $d\phi/dy$, as shown in Fig. 2f. The choice of each meta-atom is based on the criterion of possessing $T > 0.9$ as well as its corresponding encoded $\phi$. The representative meta-atoms are visually depicted in Fig. 2g and showcase clear phase delays, which enable the propagation of highly sound-transparent waves in pressure fields with plane-wave generation on the left. Additionally, our meta-atoms benefit from angle-invariant characteristics that guarantee the same functionality under oblique illumination (see details in Supplementary Note 2 and 5).

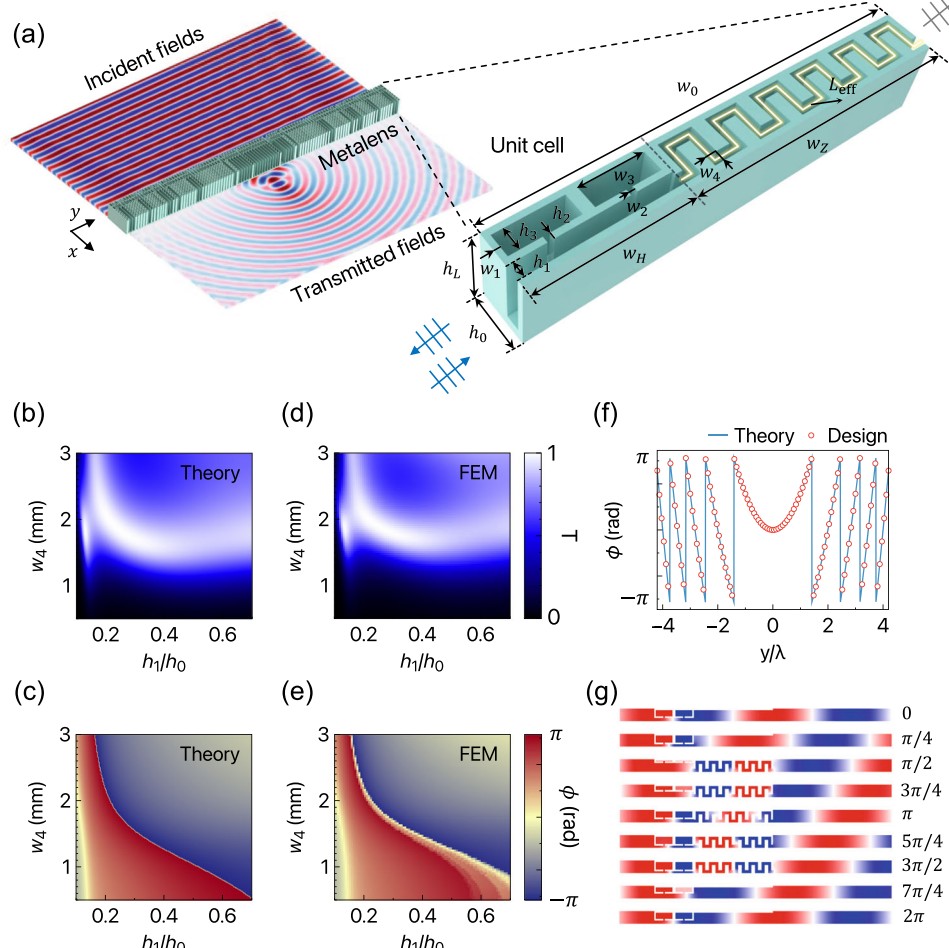

**Fig. 2 | Design of the WFOH metalens. a** Schematic of focusing field realized by the metalens and the structure of a single unit cell. The operating wavelength is $\lambda = 100$ mm (3.43 kHz). **b, c** Theoretically and (**d, e**) numerically calculated transmission and phase of unit cells with respect to the geometrical parameters $h_1$ and $w_4$. **f** Continuous (theory) and discrete (design) phases along the $y$-axis. **g** Representative meta-atoms with phase delays and highly transparent waves in pressure fields.

## Experimental realization of wide field-of-hearing metalens

A schematic of the experimental setup used to demonstrate the characteristics of the proposed WFOH metalens is depicted in Fig. 3a. The measurements are conducted in an acoustic chamber to eliminate unwanted echoes from the boundaries and background noise. We fabricate the prototype metalens using photo-polymer ultraviolet (UV) resin and the stereolithography apparatus (SLA) method (see details in Methods). Figure 3b shows a photographic image of a partial section of the manufactured metalens. Our proposed metalens is not limited to a specific working frequency; however, 3.43 kHz (corresponding to a wavelength of $\lambda = 100$ mm in air) has been selected for the sample-fabrication capability of the 3D-printing apparatus fitted in our setup size.

In the experiments, the plane waves are generated by a loudspeaker array, and the metalens is placed 100 cm away from the source. The spatial intensity distribution is measured using four 1/4-inch microphones (Type 4957; Brüel & Kjær) positioned in holes on the top plate of the acoustic chamber. These microphones scan the region near the target focal point using point-by-point measurements (see Supplementary Note 6). The size of the measuring area is $30 \times 45$ cm² , and each pixel size is $1 \times 1$ cm² ($0.1\lambda \times 0.1\lambda$). We measure the distributions of sound intensity at different AOIs ($\theta_i = 0°$, $10°, ..., 60°$) while tilting the metalens instead of the loudspeaker array. Figure 3d shows the measured intensity distributions of the WFOH metalens at AOIs of 0°, 20°, 40°, and 60°. Additional measured results

(at AOIs of $\theta_i = 10°$, $30°$, and $50°$) can be found in Supplementary Note 7. Excellent agreement is observed between the simulated and measured results (Fig. 3c, d), demonstrating the aberration-free focusing fields even at large AOIs. These robust results are in contrast to the intensity distributions with large aberrations obtained for conventional metalenses, shown in Supplementary Note 8.

Indeed, our proposed WFOH metalens is more versatile than the conventional metalens owing to the miniature acoustic planar platform (Fig. 4a). Its high NA and relatively short focal length ensure a small form factor (short distance of the aperture-to-hearing plane and minimized reception area) while maintaining focusing capability across a wide range of AOIs. We further quantify the WFOH characteristics. In the left panel of Fig. 4b, PSFs of the WFOH metalens obtained by normalizing line intensity on the focal plane exhibit excellent agreement between the theoretical, FEM, and experimental results, emphasizing the accuracy and reliability of the proposed design. It is worth noting that the decreasing trend (gray dashed line and asterisk) in intensity across the AOIs is mainly due to the effect of the cosine-dependent relative illumination (vignette effect). Additionally, we calculate the peak signal-to-noise ratio (PSNR) to quantify the focusing quality[68–70], obtaining an average PSNR of 20.32 dB, indicating high-fidelity sound capturing performance (see details in Supplementary Note 9). On the other hand, as shown in the right panel of Fig. 4b, the PSFs indicate that the conventional metalens only works effectively at normal incidence. Under oblique incidence, the actual

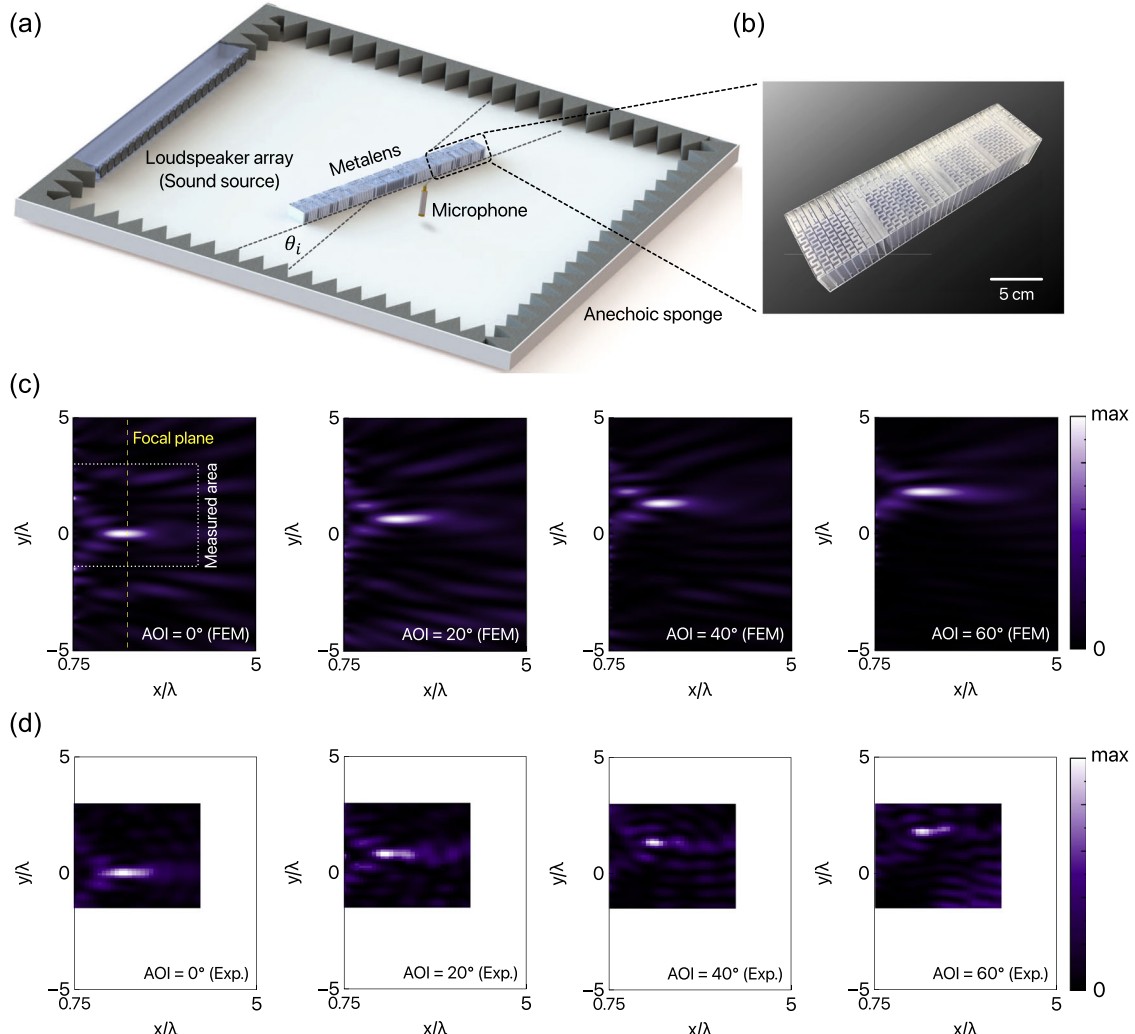

**Fig. 3 | Experimental realization of the WFOH metalens. a** Schematic of the experimental setup. The size of the acoustic chamber is $200 \times 200 \times 6$ cm³. **b** Photograph of the fabricated sample (partial section). **c** Simulated and (**d**) experimentally measured ($30 \times 45$ cm² rectangular region) acoustic intensity distributions of the WFOH metalens at 0°, 20°, 40°, and 60° in AOIs.

focal spots deviate significantly from the desired focal spot, denoted by pink vertical bars following $f \tan\theta_i$, accompanied by considerable intensity suppression due to large off-axis aberrations.

Measuring the position of focal spots, which can be predictably located along AOIs, is essential for source-tracking. We observe that the measured focal positions of the WFOH metalens are in good agreement with the theoretical $f \sin\theta_i$ curve, which remains valid up to an AOI of 70° (Fig. 4c). One can quantify the focusing efficiency[21,24] defined by $\eta = \int_{-y}^{y} I|_{x=f}\, dy / \int_{-D/2}^{D/2} I_0|_{x=0}\, dy$ integrated over the $[-y, y] = 3 \times$ FWHM region at the focal spot for each different AOI, where $I_0$ is the incident intensity and $I$ is the focused intensity. As shown in Fig. 4d, we observe that $\eta$ in the conventional metalens is no longer constant and exhibits a sharp decrease across the AOIs owing to strong off-axis aberrations. However, the WFOH metalens results in a flat-like $\eta$ up to an AOI of 70°, which is a signature of its wide-angle focusing capability. Note that the moderate focusing efficiency of the WFOH metalens originates from the intrinsic effective aperture effect ($D_{\text{eff}} = 2f$).

## Discussion

In summary, we introduce a metric, FOH–previously overlooked in existing acoustic metalenses–and experimentally demonstrate the WFOH capability. Our proposed metalens is designed to capture and

focus sound while reducing off-axis aberrations such as coma and field curvature. This is achieved through perfect acoustic symmetry conversion, allowing for WFOH capability across an extensive angular range of approximately 140° without the need for the angle-dispersive configuration typically required in conventional metalenses. Furthermore, the planar receiving plane and the short aperture-to-hearing plane distance facilitate compact wide-angle sound capture. Although this feature is realized for high-fidelity source-tracking in the audible range, it also offers potential extensions to its applicability in ultrasonic and submerged environments. The versatility and small form factor of an acoustic single-layer metalens, combined with its WFOH capability, make it well-suited for various scenarios where efficient sound capture, imaging capabilities, and wireless communication are required. The exploration of high-frequency regimes, particularly within the medical field, is of great interest. It involves addressing impedance mismatch and resolving the trade-off between the boundary layer effect and meta-atom period (see Supplementary Note 10 for details). Moreover, an improved FOH capability through the use of bianisotropy and non-locality with various meta-atom designs is highly expected (see Supplementary Note 11 for details). While our current experimental demonstration is focused on one-dimensional audible sound focusing, extending this into two and three dimensions holds promise for exploring novel applications. We believe

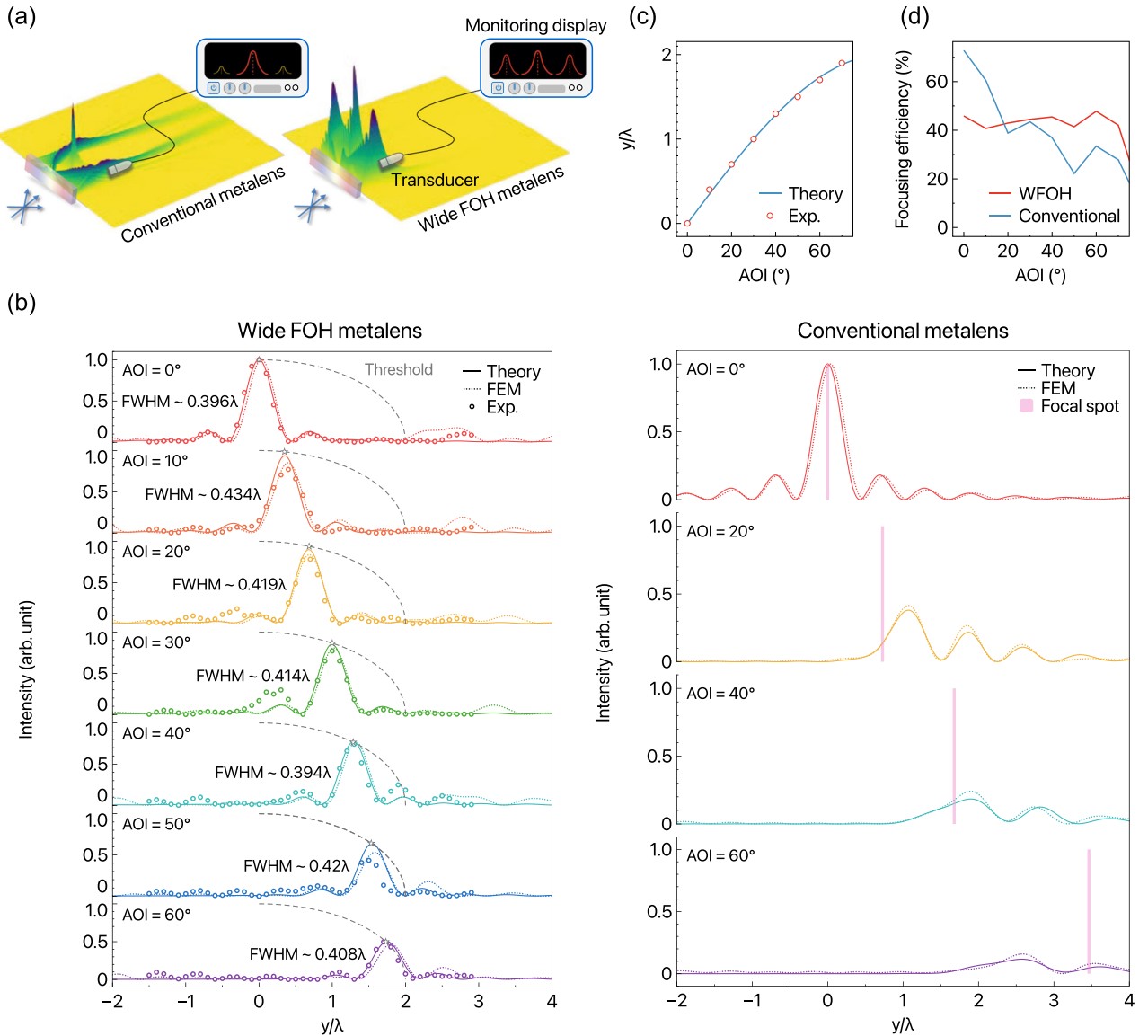

**Fig. 4 | Demonstration of the wide-angle sound-reception system. a** Schematic of sound monitoring with conventional and WFOH metalenses. **b** Sound intensity distributions of the WFOH (left panel) and conventional metalens (right) on the receiving focal plane, ranging from 0° to 60° in AOIs; measured data–dots (only for the left panel); theoretical results–solid lines and FEM results–dotted lines. In the left panel, dashed gray lines labeled 'Threshold' represent the cosine-dependent

relative illumination effect. Vertical bars in the right panel indicate the desired focal spot of the conventional metalens, $f \tan \theta_i$. **c** Transverse focal-spot location of the WFOH metalens as a function of AOI. The measured points are shown by dots, and the solid line indicates the theoretical $f \sin \theta_i$ curve. **d** Calculated focusing efficiencies of WFOH and conventional metalenses as a function of AOI.

that the proposed work lays the groundwork for further advancements in acoustic metalens technology and engineering, thereby enabling prospective opportunities in the field of WFOH sound reception. This strategy can alleviate the necessity for complex and high-powered electronic components, thus facilitating practical and efficient implementations. We also note that the outlook of the WFOH towards wideband and achromatic characteristics remains an open-ended question.

## Methods

### Details on symmetry conversion towards WFOH metalenses

For achieving the perfect conversion from rotational symmetry in spherically symmetric gradient-index lenses to translational symmetry in planar metalenses, in a two-dimensional form, one can express

$$\phi(y + \Delta_y, z + \Delta_z) = \phi(y, z) - (k_0 \sin\theta_y y + k_0 \sin\theta_z z), \quad (3)$$

where $\phi(y, z)$ is the phase distribution encoded in the metalens, and $\Delta$ represents a corresponding translational shift. This leads to

$$\frac{\partial \phi}{\partial y} = -\frac{k_y}{\Delta_y} y, \quad \frac{\partial \phi}{\partial z} = -\frac{k_z}{\Delta_z} z, \quad (4)$$

where $k_y = k_0 \sin\theta_y$ and $k_z = k_0 \sin\theta_z$. By integrating both sides of Eq. (4) we have a two-dimensional phase profile as follows:

$$\phi(y, z) = -\frac{1}{2}\left(\frac{k_y}{\Delta_y} y^2 + \frac{k_z}{\Delta_z} z^2\right). \quad (5)$$

Considering the spherical symmetry ($\Delta_y = \Delta_z = \Delta$, and $k_y = k_z = k_0 \sin\theta_i$) yields,

$$\phi(y, z) = -\frac{k_0}{2}\left(y^2 + z^2\right) h(\theta_i), \quad (6)$$

where $h(\theta_i) = \sin\theta / \Delta$. Now we consider translational symmetry which is effectively converted from rotational symmetry of a spherically

symmetric lens i.e., $\Delta = f \sin \theta_i$, and a resulting quadratic phase form can be established for a planar lens as follows:

$$\phi(r) = -\frac{k_0}{2f} r^2, \tag{7}$$

where $r = \sqrt{y^2 + z^2}$. We also note that Eq. (7) can be obtained through a second-order (quadratic) Taylor expansion of the hyperbolic phase profile. For oblique illuminations in the $xy$-plane with an arbitrary $\theta_i$ to the normal axis of the lens, the effective phase distribution at the EP can be expressed as

$$
\begin{aligned}
\phi_{\mathrm{EP}}(r, \theta_i) &= -\frac{k_0}{2f} r^2 + \phi_{\mathrm{PS}}, \\
&= -\frac{k_0}{2f}[(y + f\sin\theta_i)^2 + z^2] + \frac{fk_0\sin^2\theta_i}{2},
\end{aligned} \tag{8}
$$

where $\phi_{\mathrm{PS}} = -k_0 y \sin\theta_i$ and the focal-point position is determined by the quantity $f\sin\theta_i$. If we consider a one-dimensional lens, then the same formula is obtained as presented in Eq. (2). See details in Supplementary Note 2 for further characterization of the effective phase distribution.

## Numerical simulation

Throughout this work, full-wave simulations were conducted using a finite-element method based on commercial software, COMSOL MULTIPHYSICS 6.1 with the "pressure acoustics module". The mass density and sound speed of the background medium (air) are 1.21 $\mathrm{kg \cdot m^{-3}}$ and 343 $\mathrm{m \cdot s^{-1}}$, respectively. The base material for lens fabrication is photo-polymer UV resin with a mass density of 1170 $\mathrm{kg \cdot m^{-3}}$ and sound speed of 2700 $\mathrm{m \cdot s^{-1}}$. Its acoustic impedance is significantly greater than that of the background medium; thus, the structure is considered to be acoustically rigid.

## Experiments

All samples were fabricated using SLA-type 3D printing (0.1 mm resolution). The metalens sample was divided into three equal sections without affecting the experimental results. Each piece was manufactured under identical printing conditions and then combined together. All experiments were performed in a two-dimensional waveguide chamber. A loudspeaker array with 22 speaker units (TC5FB00-08; Peerless) was used as a plane-wave source. Each speaker unit was equally arranged at an interval of 8 mm, which is sufficiently small compared to the working wavelength. The input signal was applied to the loudspeaker array using a function generator (33220A; Keysight) and power amplifier (HSA 4052; NF Corporation). Four calibrated microphones (1/4″ Type 4957; Brüel & Kjær) and a multichannel data-acquisition module (Pulse Type 3676; Brüel & Kjær) with the commercial Labshop software (Brüel & Kjær) were used to measure the acoustic signals. Further details on the experimental setup and plane-wave propagation in free space are provided in Supplementary Note 6.

## Data availability

Data that support the findings of this study are available from the corresponding authors upon request.

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

## Acknowledgements

This work was financially supported by the POSCO-POSTECH-RIST Convergence Research Center program funded by POSCO, the National Research Foundation (NRF) grants (NRF-2019R1A5A8080290, NRF-2022M3C1A3081312, RS-2023-00283667) funded by the Ministry of Science and ICT (MSIT) of the Korean government, the Korea Evaluation Institute of Industrial Technology (KEIT) grant (No. 1415179744/20019169, Alchemist project) funded by the Ministry of Trade, Industry and Energy (MOTIE) of the Korean government, and the grant (PES4400) from the endowment project of "Development of smart sensor technology for underwater environment monitoring" funded by Korea Research Institute of Ships & Ocean engineering (KRISO).

## Author contributions

J.R., D.L., and B.O. conceived the idea and initiated the project. D.L. and B.O. performed theoretical analyses and numerical simulations. B.O., D.L., and J.P. performed the experiments. S.K. supported the experiments. S.M. and K.S. supported the data analyses. All authors participated in discussions and confirmed the final manuscript. J.R. guided the entire work.

## Competing interests

The authors declare no competing interests.
