## [Peer Review File · Nature Communications]

Wide field-of-hearing metalens for aberration-free sound captureREVIEWER COMMENTS

Reviewer #1 (Remarks to the Author):

In this manuscript, the authors construct a composite unit composed of zigzag channels and Helmholtz resonators units to achieve arbitrary regulation of transmission amplitude and phase. And by arranging the composite metasurface units according to a certain phase gradient, a type of metalens is formed, achieving a wide FOH effect similar to optical wide FOV in the waveguide. There is a certain academic innovation in the work, but metasurface units with similar functions (i.e. phase amplitude arbitrary regulation) have been reported around 2018 in APL, only they have not been used to achieve FOH functions. Therefore, the reviewer believe that the physics presented in this manuscript is not new enough to satisfy the standards of NC. The detail comments are as follows:

1. Although wide FOH has not been reported, the author still constructed metalens using metasurface units without introducing new physical principles. The authors seem to provide a physical relationship that needs to be satisfied to implement wide FOH, rather than breaking through anything.
2. For transmissive-type metasurfaces, even if only one type unit (zigzag channels unit or Helmholtz resonators unit) is used, the loss is very large, let alone when two units are connected in series. High loss can seriously hinder the engineering application value of metalens.
3. Although the focal spot implemented by the authors is subwavelength, the size of the entire metalens is many times that of the wavelength, and it can only work in waveguide environments, lacking engineering application value.
4. The implementation of related concepts in airborne has a greatly reduced significance, while it has a greater significance in ultrasound.

Reviewer #2 (Remarks to the Author):

This paper presents a design of a wide field of view acoustic metasurface, which can perform efficient acoustic focussing for a wide range of incident angles. The results appear quite interesting, however the presentation and explanation of the work is unclear, and in places appears contradictory.

The authors start by noting that most metasurface lens designs consider the hyperbolic phase distribution. They then show how the ideal phase profile, which depends on the incident angle, has a relatively complex mathematical expression, and is not easy to implement in a realistic metamaterial structure. For this reason they introduce the quadratic phase profile (previously demonstrated for optical metasurfaces), which has a simpler dependence on the incident field.

To back up this claim that the quadratic phase profile is simpler, the authors refer to supplementary figure S1. However, it's not at all clear why Figure S1(b) should be considered a "flattened and undistorted phase distribution" when visually it appears much more complex than Figure S1(a) for the ideal phase profile.

Furthermore, the quadratic phase profile as presented in Equation (1) appears to be angle dependent - yet in Figure S4, the authors demonstrate that (to a good approximation) their meta-atoms are angle-independent.

I am able to clarify my understand by referring to the cited papers on optical wide field of view metalenses. These seem to indicate that the authors here are confusing the designed phase profile of the lens with the phase of the transmitted light in Eq. (1). So this confusion needs to be cleared up, and the manuscript made sufficiently self-contained that its key points can be understood, including by acoustics practitioners who may not be familiar with the work in optics.

Figure 1 is intended to present the key ideas to the readers, but it does not do this job well. Subfigure

(a) seems to contain nothing useful that is not shown more clearly in subfigure (b). Subfigure (c) shows the Fourier components of the point-spread function - it's unclear what useful information this conveys compared to just showing the point spread function. Finally, subfigure (d) appears to be just a plot of the well-known relationship between focal length, aperture size and numerical aperture, which does not seem to warrant an entire figure.

The authors mention several times the "translational symmetry" of their design. This is not explained well, but would make sense once Eq (1) is properly explained.

In the supplementary material, the authors show that their meander line structures can only achieve full transmission when their transmission phase is 0 - due to the use of Fabry-Perot resonances. For this reason additional Helmholtz resonators are introduced - it's unclear whether the meander lines are really necessary, or if the Helmholtz resonators would be sufficient on their own (as they have been shown in other designs - although typically 3 or more Helmholtz resonators are required in such cases).

Note that the claim on page 4 that the Helmholtz resonators "amplify the signal" is misleading - since these are passive structures. Possibly the authors are referring to impedance matching with the Helmholtz resonators.

On page 5 the authors mention "simultaneous modulation of T and Phi" - yet it seems that they only require structures with (approximately) unity transmission, so only modulation of Phi is relevant to this work.

The Experimental realization section is the strongest part of this paper - the results clearly show good performance, the agreement between experiments and numerics is fairly good, and the discussion/explanation is easy to follow.

In the abstract and discussion section, it is claimed that this is an "ultrathin and highly compact design" - but based on page 5 it is stated that $w_0 = 0.75 \cdot \lambda$, which is comparable to the wavelength. On the other hand, $h_0 = \lambda / 12.5$ is relatively small - this is essential for achieving the wide field of hearing.

To understand the significance of the achieved results, it is necessary for the reader to look at Supplementary Figure S7. I think it would be much clearer if sub-figure S7(e) was somehow incorporated into the main text.

In Figure 1 caption and on page 4 of the main text, the term "perfect conversion from rotational symmetry to translational symmetry" is used, without proper explanation. Either this concept should be explained properly in the main text, or just removed entirely.

In Figure 1(c), it is confusing to use "contrast" for a quantity when the term "MTF" is used in the main text - apparently referring to the same thing.

In Equation (1), the term x should be removed, since only the 1D case is considered in all other equations.

In Figure 4, the line labeled "Threshold" should be better explained in the caption (it is mentioned in the main text, but with a different name)

Overall this paper shows interesting experimental results, but the accompanying theoretical analysis and explanation need to be made much clearer.

Reviewer #3 (Remarks to the Author):

In this study, the authors proposed an acoustic metalens exhibiting wide field-of-hearing (FOH), which is an acoustical analogy of the wide field-of-view (FOV) in optics. The acoustic metalens is composed of several unit structures with different phase shifts to satisfy the phase distribution for wide FOH. Each unit structure, constructed from Helmholtz resonators with zigzag channels, exhibited high transmission and various phase shifts from 0 to 2π . The phase shift and transmission of the unit structures were calculated using a theoretical model derived by the authors and the results were validated by FEM simulations. The characteristics of the wide FOH metalens also were verified by experiments. Despite aforementioned efforts, the reviewer is not convinced of significant contribution in this work for the following reasons.

(1) The key to the wide FOH is the specific phase distribution discovered already in terms of FOV by Pu et al. [Opt. Express, 25, 31471–31477 (2017)]. In the manuscript, the phase distribution was employed without any modification or extension when bringing it from the optical to the acoustical domain.

(2) Besides, discussions for the underlying physics of wide FOH in the supplementary materials were almost the same with those in a recent review paper [Nanophotonics, 11(1), 1-20 (2022)]. It seems that 'phase surface contours' presented in the supplementary materials were similar to those in the Nanophotonics paper with only the transformation of the x-axis from $\sin\theta$ to θ : nevertheless, this paper was not referred in both the manuscript and supplementary materials.

(3) Regarding the implementation of the wide FOH, acoustic metalenses in a paper [J. Phys. D, 52, 385303 (2019)] also showed that the wide angles of incidence could be allowed to 60 degrees, but the manuscript did not provide a concrete comparison with this or any other related works. As for the unit structure, the width of the unit structure in the manuscript was thinner than that in the J. Phys. D, but one in J. Phys. D also could be sufficiently thin to satisfy the rapid phase gradient necessary for implementing a wide FOH metalens when considering the results for the wide angles of incidence as mentioned above.

Based on the concerns raised regarding novelty and significant improvement over prior works, the reviewer cannot recommend the publication of the present manuscript in Nature Communications.

Response to Reviewers' Comments

(NCOMMS-23-37112)

Title:

Wide field-of-hearing metalens for aberration-free sound capture

Authors:

Dongwoo Lee*, Beomseok Oh*, Jeonghoon Park, Seong-Won Moon, Kilsoo Shin, Sea-Moon Kim, and Junsuk Rho†

Overall remarks:

Dear Reviewers,

We would like to express our sincere gratitude to the referees for their constructive comments and insights on improving the quality and presentation of our paper. We have made changes in response to your valuable feedback for the resubmission to "*Nature Communications*". All the points raised by the reviewers have been incorporated into the revised manuscript. "Responses" and "modifications" made in this revision are colored in "blue" and "orange", respectively.

We thank the reviewers again for dedicating their time to a thorough review of our manuscript.

Response to Reviewer #1's comments:

Comment: In this manuscript, the authors construct a composite unit composed of zigzag channels and Helmholtz resonators units to achieve arbitrary regulation of transmission amplitude and phase. And by arranging the composite metasurface units according to a certain phase gradient, a type of metalens is formed, achieving a wide FOH effect similar to optical wide FOV in the waveguide. There is a certain academic innovation in the work, but metasurface units with similar functions (i.e. phase amplitude arbitrary regulation) have been reported around 2018 in APL, only they have not been used to achieve FOH functions. Therefore, the reviewer believe that the physics presented in this manuscript is not new enough to satisfy the standards of NC. The detail comments are as follows:

Reply:

We would like to thank the reviewer for his/her careful review and valuable suggestions, which have helped to improve the manuscript. We are delighted to observe that Reviewer #1 acknowledges the academic innovation in some aspects of our work. In the following, we strive to provide clear responses to the comments, along with our best efforts in proposing substantial modifications to the manuscript. We hope that our point-by-point response meets your standards.

We agree that the phase and amplitude modulation (PAM) method has been previously reported [R1-R3]. According to the reviewer's concern about our original statement about PAM, '*These findings can be further utilized for applications such as acoustic holography and particle manipulation*', we wish to clarify that our intention was not to highlight new findings but to describe the potential capability of achieving PAM using our new meta-atom configuration (HRs with a zigzag channel). Recognizing a possible misunderstanding in the original statement, we have made a moderate adjustment accordingly and incorporated additional references related to PAM in the revised manuscript.

Moreover, we are afraid that the emphasis of our work is somehow underestimated based on the reviewer's perspective regarding the statement "*only they have not been used to achieve FOH functions*". A detailed response regarding the novelty and physical aspects of our work will be given to proper space (please refer to "Reply 1") for a comprehensive explanation in the following.

Change:

1. Please refer to lines 162-163 on page 5 of the revised manuscript.

"..., we can obtain the simultaneous modulation of T and ϕ ⁶⁴⁻⁶⁶ by adjusting geometric parameters w_4 and h_1 ;"

2. Please refer to lines 164-168 on page 5 of the revised manuscript.

“We note that our meta-atoms can also find utility in applications such as acoustic holography and particle manipulation, which require independent control of T and ϕ . However, for the purpose of the present study, we exclusively focus on phase modulation to achieve exceptionally efficient focusing over a wide range of incident angles.”

[64] Tian, Y., Wei, Q., Cheng, Y. & Liu, X. Acoustic holography based on composite metasurface with decoupled modulation of phase and amplitude. *App. Phys. Lett* 110, 191901 (2017).

[65] Ghaffarivardavagh, R., Nikolajczyk, J., Glynn Holt, R., Anderson, S. & Zhang, X. Horn-like space-coiling metamaterials toward simultaneous phase and amplitude modulation. *Nat. Commun.* 9, 1349 (2018).

[66] Zhu, Y. et al. Fine manipulation of sound via lossy metamaterials with independent and arbitrary reflection amplitude and phase. *Nat. Commun.* 9, 1632 (2018).

[R1] Tian, Y., Wei, Q., Cheng, Y. & Liu, X. Acoustic holography based on composite metasurface with decoupled modulation of phase and amplitude. *App. Phys. Lett* 110, 191901 (2017).

[R2] Ghaffarivardavagh, R., Nikolajczyk, J., Glynn Holt, R., Anderson, S. & Zhang, X. Horn-like space-coiling metamaterials toward simultaneous phase and amplitude modulation. *Nat. Commun.* 9, 1349 (2018).

[R3] Zhu, Y. et al. Fine manipulation of sound via lossy metamaterials with independent and arbitrary reflection amplitude and phase. *Nat. Commun.* 9, 1632 (2018).

Comment 1:

Although wide FOH has not been reported, the author still constructed metalens using metasurface units without introducing new physical principles. The authors seem to provide a physical relationship that needs to be satisfied to implement wide FOH, rather than breaking through anything.

Reply 1:

We appreciate the reviewer’s comment on the novelty and physics aspects of our work. We agree that our strategy draws inspiration from the wide field-of-view (FOV) which has recently garnered attention in the optical domain. Our primary focus is indeed on introducing the concept of *field-of-hearing (FOH)* that goes beyond the narrow limited FOH capability observed in previous studies. In this context, we believe our work is novel and will provide high impact on the acoustic metalens community by providing a comprehensive understanding of quadratic phase as well as design principles that facilitate the characterization of lenses’ performance. The details are as follows:

Although metasurface-based acoustic focusing has been extensively studied, there has been no prior exploration, approach, and methodology specifically addressing FOH. The FOH capability, as a new functionality with a distinct degree of freedom, remains to be further investigated ahead. Moreover, none of the previous studies made efforts to characterize lens performance (such as NA, aberrations, MTF, focusing efficiency, and more). Rather, they mainly provided proof-of-concept demonstrations for sound focusing without adhering to relevant considerations for characterizing the lens's performance. Building upon our claim of the new introduction of the wide FOH capability, we firmly believe that our present work will make substantial contributions and guide the development of acoustic metalenses.

For more details, we conducted a comprehensive literature survey on acoustic metalenses, incorporating [S16, the paper introduced by Reviewer #3], now included in the revised supplementary information. This table provides evidence of the novelty in our present work, particularly for the FOH capability as well as the lens's characteristics.

To address and back up the physics aspects of concern raised by the reviewer, we have reinforced our manuscript by adding detailed explanations for the quadratic phase and its symmetry conversion. Additionally, we have incorporated comprehensive insights into the working principle, specifically providing the investigation of actual phase distributions at exit pupils, detailed in Supplementary Note 2. To the best of our knowledge, exploring the actual phase distributions to explain symmetry conversion has not been undertaken previously in the optical domain. We believe that explanations based on the actual phase distribution offer a clearer understanding of symmetry transformation, especially for acoustic practitioners, facilitating a more intuitive comprehension. Respectfully, we wish to invite the reviewer for a second round of consideration, highlighting the following key points. In essence, our present work strengthens its novelty and meets NC standards through the following primary aspects:

- 1) First introduction of a new metric 'FOH'—previously overlooked in acoustic metalenses—and highlighting its necessity, along with the experimental realization of unprecedented WFOH capability.
- 2) Comprehensive characterization of a lens performance from a sensing point-of-view: covering key measures such as NA, aberrations, MTF, focal points tracking, and focusing efficiency.
- 3) Comprehensive interpretation on the quadratic phase and symmetry conversion mechanism with considerations of actual phase distributions.

Given the discussions above, we anticipate that our work will serve as a significant milestone by newly introducing the notion of FOH in acoustic metalens technology, paving the way for new horizons to be explored further. We hope that the reviewer may recognize our efforts and find our work an intriguing contribution to the community.

Change 1:

1. Please refer to lines 95-114 on page 4 of the revised manuscript.

“The WFOH system exploits the quadratic form of the one-dimensional phase profile, which is expressed as follows:

$$\phi(y) = -\frac{k_0}{2f}y^2, \quad (1)$$

where y represents the transverse position across the lens. The primary concept of Eq. (1) involves the transformation of a spherically symmetric gradient-index (Luneberg) lens in to a *flat-configuration* where both the lens’s geometrical shape and the focal plane can be flattened—achieving a perfect conversion from rotational symmetry to translational symmetry (Fig. 1(a)). If incident sound waves are projected in the xy -plane at an arbitrary AOI, the effective phase shift $\phi_{\text{PS}} = -k_0 y \sin \theta_i$ is incorporated, yielding the following phase distribution after transmitting through the metalens or at the exit pupil (EP) of the metalens:

$$\phi_{\text{EP}}(y, \theta_i) = -\frac{k_0}{2f}y^2 + \phi_{\text{PS}} = -\frac{k_0}{2f}(y + f \sin \theta_i)^2 + \frac{f k_0 \sin^2 \theta_i}{2}. \quad (2)$$

The effect of oblique incidence is evident in the transverse translation of the focal position by $f \sin \theta_i$, in Eq. (2). In the case of spherically symmetric lenses, the focal plane takes a non-planar form, requiring complex-receiving components; otherwise, it leads to field curvature aberration based on the planar focal plane. However, through a perfected symmetry conversion process, WFOH metalenses possess a planar receiving plane with predictable focal positions, enabling straightforward wide-angle acoustic sensing (Fig. 1(a)). Details on the derivation of the quadratic phase profile are provided in Methods. We emphasize that Eq. (2) does not imply the necessity for meta-atoms to possess angular dispersion, i.e., angle-dependent phase delay and transmission. Rather, it denotes the effective transformation from the self-contained phase shift under oblique illumination scenarios (see Supplementary Note 2 for a detailed discussion). Moreover, these characteristics enable successful realization comparable to a Fourier lens, applicable to diverse uses such as spatial filtering and compressed sensing⁶², all within a compact planar form factor.”

2. Please refer to page S3-S4 of the supplementary information (Supplementary Note 2, which is newly added).

“Supplementary Note 2: Wide-angle responses of WFOH metalenses by symmetry conversion

In this section, we delve into more details regarding the symmetry conversion based on the actual

phase distributions at the exit pupils of metalenses.

As discussed in the main text, the WFOH metalens is designed with Eq. (1). The ability to achieve wide-angle focusing with angle-dispersion-free meta-atoms is attributed to the inherent characteristics of the quadratic phase. Specifically, it is owing to the effective phase shift ($\phi_{PS} = -k_0 y \sin \theta_i$) and its symmetry conversion property, which is induced by the oblique illumination. For direct investigation about this, we show the actual phase distributions at the exit pupils of metalenses, i.e., phases of the transmitted waves (Fig. S2).

Fig. S2(a) represents the case of the WFOH metalens, where the blue and red markers denote the results of FEM and experiment, respectively. Also, the shaded area indicates the effective aperture area, $[-f + f \sin \theta_i, f + f \sin \theta_i]$ (further discussed below). We can clearly observe an effective transverse shift in the phase distribution with different AOIs, despite the passive configuration of the designed metalens (see also the angle-invariant characteristics of meta-atoms, as shown in Fig. S5). Despite slight discrepancies from the theory in both FEM and experimental results, we can see overall similar trends. We note that various factors may contribute to deviations in the phase distribution of a real metalens from its ideal counterpart [S3], e.g., non-uniform transmittance of meta-atoms, and non-local effects [S4, S5] between adjacent unit cells. Fig. S2(b) represents the phase distributions of metalenses designed based on the hyperbolic phase profile. It is apparent that beyond 10 degrees, the desired phase distributions fail to form, resulting in the inability to generate the ideal wavefront and consequently leading to aberrations that hinder wide-angle focusing.

Additionally, we characterize the symmetric properties of quadratic phase profile with the generalized law of refraction, which can be expressed as follows:

$$n_t \sin \theta_t - n_i \sin \theta_i = \frac{\lambda}{2\pi} \frac{d\phi}{dy} = \frac{k_y}{k_0}. \quad (\text{S2})$$

Here, n_t (n_i) indicate the refractive indices where the sound is transmitted (incident). We only consider one background medium (air) i.e., $n_t = n_i = 1$. From Eq. (S2), we see the normalized transverse wavenumber $\frac{k_y}{k_0}$ is related to the sound refraction with metasurface-induced phase discontinuity. The normalized transverse wavenumber of the quadratic phase can be expressed by

$$\frac{k_y}{k_0} = -\left(\frac{y}{f} - \sin \theta_i\right). \quad (\text{S3})$$

The quadratic phase maintains spatial symmetry along the axis of symmetry $f \sin \theta_i$, as shown in Eq. (S3). Based on Eq. (S3) and the condition $|k_y| > k_0$ for an imaginary wavevector along

the acoustic axis ($k_x = \sqrt{k_0^2 - k_y^2}$), we obtain the effective aperture as follows:

$$\frac{|k_y|}{k_0} = \left| \frac{y - f \sin \theta_i}{f} \right| > 1,$$

$$-f + f \sin \theta_i < y < f + f \sin \theta_i. \quad (\text{S4})$$

Therefore, the effective aperture ($D_{\text{eff}} = 2f$) varies with $f \sin \theta_i$ in accordance with AOIs.”

Fig. S2. Phase distributions at the exit pupils of (a) WFOH and (b) conventional metalens; theoretical-solid lines, FEM-blue dots, and measured-red dots. Shaded area in (a) indicates the effective aperture (D_{eff}); outside this region, the wavevector becomes evanescent.

[S3] M. Zhao, M. K. Chen, Z.-P. Zhuang, Y. Zhang, A. Chen, Q. Chen, W. Liu, J. Wang, Z.-M. Chen, B. Wang, et al., *Light Sci. Appl.* 10, 52 (2021).

[S4] X. Wang, R. Dong, Y. Li, and Y. Jing, *Rep. Prog. Phys.* 86, 116501 (2023).

[S5] K. Shastri and F. Monticone, *Nat. Photonics* 17, 36 (2023).

3. Please refer to page S15-S16 of the supplementary information (Supplementary Note 9, which is newly added).

“Supplementary Note 9: Comparison with previous studies on acoustic metalenses

We compare our work with previous studies on acoustic focusing using planar metasurfaces. A detailed comparison, as shown in Table S1, highlights crucial performance metrics of the lens, particularly for the FOH. To our knowledge, the determination of the size and focal length of metalenses seems arbitrary in Refs. [S10-S25], revealing a deficiency in considering NA during the design process, often due to the predominant focus on proof-of-concept demonstrations regarding focusing capabilities. By addressing the existing limitations of acoustic metalenses in achieving WFOH while adhering to the hyperbolic phase profile (often recognized as the trade-off between high-NA and wide FOV in conventional metalenses [S26]), our current work significantly enhances wide-angle focusing capability—an aspect often overlooked in previous studies—by emphasizing the necessity of considering FOH. While Ref. [S12] presented results for oblique incidence, it lacked detailed interpretation with its narrow FOH.”

[S10] C. Kim, J. Kim, and W. Jeon, *J. Sound Vib.* 529, 116910 (2022).

[S11] W. Li, F. Meng, and X. Huang, *Appl. Phys. Lett.* 117 (2020).

[S12] P. Wang, G. Yu, Y. Li, X. Wang, and N. Wang, *New J. Phys.* 22, 023006 (2020).

[S13] F. Zhang, E. Perkins, S. Wang, G. T. Flowers, and R. N. Dean, *Appl. Phys. Express* 12, 087002 (2019).

[S14] J. Chen, J. Xiao, D. Lisevych, A. Shakouri, and Z. Fan, *Nat. Commun.* 9, 4920 (2018).

[S15] S. Tang, B. Ren, Y. Feng, J. Song, and Y. Jiang, *J. Appl. Phys.* 129 (2021).

[S16] K. Gong, X. Wang, H. Ouyang, and J. Mo, *J. Phys. D: Appl. Phys.* 52, 385303 (2019).

[S17] S. Qi and B. Assouar, *J. Appl. Phys.* 123 (2018).

[S18] N.-L. Zhang, S.-D. Zhao, H.-W. Dong, Y.-S. Wang, and C. Zhang, *Appl. Phys. Lett.* 120 (2022).

[S19] X. Jiang, Y. Li, D. Ta, and W. Wang, *Phys. Rev. B* 102, 064308 (2020).

[S20] L. Xiang, L. Jian, and H. Xinjing, *IEEE Sens. J.* 22, 13989 (2022).

[S21] Y.-F. Zhu, X.-D. Fan, B. Liang, J. Yang, J. Yang, L.-l. Yin, and J.-C. Cheng, *AIP Adv.* 6 (2016).

[S22] Y. Zhu and B. Assouar, *Phys. Rev. B* 99, 174109 (2019).

[S23] H.-W. Dong, C. Shen, S.-D. Zhao, W. Qiu, H. Zheng, C. Zhang, S. A. Cummer, Y.-S. Wang, D. Fang, and L. Cheng, *Natl. Sci. Rev.* 9, nwac030 (2022).

[S24] J.-K. Weng, Y.-F. Zhu, B. Liang, J. Yang, and J.-C. Cheng, *Appl. Phys. Express* 13, 094003 (2020).

[S25] S.-D. Zhao, A.-L. Chen, Y.-S. Wang, and C. Zhang, *Phys. Rev. Appl.* 10, 054066 (2018).

Table S1. Summary of acoustic metalenses.

Ref.	NA	FOH (deg)	Type	Exp. verification	Thickness (λ)	FWHM (λ)	Focusing efficiency (%)	Operation frequency (kHz)
This work	0.9	140	2D/ Transmissive	O	0.75	0.39-0.42	41-43	3.43
[S10]	0.76	N/A	2D/ Transmissive	O	0.5	0.56	N/A	2.5, 4, 5.5
[S11]	0.74	N/A	2D/ Transmissive	O	0.54	0.52	41.5	3.1
[S12]	0.51	20	2D/ Reflective	O	0.5	N/A	N/A	2.8-5.6
[S13]	0.49	N/A	2D/ Transmissive	O	0.87	N/A	N/A	7.5
[S14]	0.5	N/A	3D/ Transmissive	O	0.05	0.98	N/A	3.43
[S15]	0.68	N/A	2D/ Transmissive	O	0.5	N/A	N/A	2.7, 4.1, 5.5
[S16]	0.8, 0.94	N/A	2D/ Transmissive	X	0.65	0.52, 0.39	N/A	5
[S17]	0.68, 0.73	N/A	3D/ Reflective	X	0.07	0.76, 0.74	N/A	3.43
[S18]	0.7	N/A	2D/ Reflective	O	0.2	0.64-0.75	N/A	3, 3.5, 4.5
[S19]	0.15	N/A	2D/ Reflective	X	0.25	0.62	N/A	500 (water)
[S20]	0.87	N/A	3D/ Transmissive	O	0.98	0.55-0.83	N/A	8
[S21]	0.86	N/A	2D/ Reflective	X	0.5	N/A	N/A	1.7, 3.4, 5.1
[S22]	0.78	N/A	2D/ Reflective	O	0.21	N/A	N/A	2
[S23]	0.79	N/A	2D/ Transmissive	O	0.87	0.59-0.7	N/A	1-4
[S24]	0.68	N/A	2D/ Reflective	X	0.25	N/A	N/A	1.3, 1.9, 2.4
[S25]	0.88	N/A	3D/ Transmissive	O	0.8	\sim 0.6	N/A	5.5

Comment 2:

For transmissive-type metasurfaces, even if only one type unit (zigzag) channels unit or Helmholtz resonators unit) is used, the loss is very large, let alone when two units are connected in series. High loss can seriously hinder the engineering application value of metalens.

Reply 2:

We thank you for this point. It is well known that visco-thermal effects become significant when the channel width is about two orders of magnitude larger than the viscous and thermal boundary layers [R4, R5]. Viscous and thermal boundary layers are defined by $\delta_v = \sqrt{2\mu/\omega\rho_0}$, $\delta_t = \delta_v/\sqrt{\text{Pr}}$ where μ is the dynamic viscosity, ω is the angular frequency, ρ_0 denotes the mass density of the medium, and Pr is Prandtl number [R4-R7]. At 3.43 kHz in air, δ_v is around 32 μm . We analyzed the transmission of meta-atoms, considering thermoviscous effects. In Fig. R1(a), blue and red lines indicate the calculated transmission without loss and with loss, respectively. The average transmission in the lossless case is 0.95, and 0.79 in the presence of thermoviscous loss. Although the loss is incorporated, we still observe that the majority of meta-atoms exhibit $T > 0.7$. Additionally, as shown in Fig. R1(b), we calculate the ratio between the viscous boundary layer and the channel width ($\Gamma = \delta_v/w$). For most meta-atoms, $\Gamma < 0.01$, indicating w is two orders of magnitude smaller than the boundary layer thickness. Consequently, the impact of thermoviscous loss in our system is not significant enough to hinder efficient wave focusing.

In principle, there exists a trade-off between the spatial spacing (or period, denoted as $p = h_0$ in our work) of discretized meta-atoms and thermoviscous loss. When p decreases, the metalens may operate closer to a continuous phase, potentially leading to small aberrations and high focusing efficiency. However, the influence of thermoviscous effects comes into play, with the potential to decrease efficiency because of lower transmission. Conversely, increasing p alleviates thermoviscous loss, yet the larger p induces parasitic diffraction, which can lead to decreased focusing efficiency. While our current design approach for the WFOH metalens in this work does not extensively consider thermoviscous effects, it is possible to identify an optimal meta-atom period that maximizes focusing efficiency once the loss effect is incorporated into the design. This aspect could be explored in our subsequent work related to ultrasonic regime.

Moreover, as indicated by the reviewer, high loss can seriously hinder the engineering application value of a metalens because it leads to a lower intensity of PSF. Absolutely, despite thermoviscous loss, our experimentally verified PSFs exhibit high contrast (Fig. 4), indicating sufficient signal-to-noise ratio (SNR), and thus acoustic signals can be clearly resolved. Moreover, expanding to a 2D WFOH metalens can allow for even higher intensity in PSFs compared to a 1D WFOH metalens, along with enhanced SNR while maintaining a minimized receiving plane. In practical applications, it is notable that acoustic signals used for sensing purposes tend to be

weak. Conventional electrical acoustic sensor devices, such as phased array transducers, often struggle to detect these weak signals effectively. Nonetheless, metalenses harness the benefits of traditional lenses, offering high sensitivity with cost-effective and compact configurations. In this context, we strongly believe that integrating WFOH metalenses with MEMS microphone arrays may hold immense potential for next-generation acoustic camera systems with high sensitivity and enhanced resolution.

Fig. R1. (a) Transmission of meta-atoms without loss (blue) and with thermoviscous loss (red). Horizontal dashed gray line indicates $T = 0.7$. (b) Ratio between the viscous boundary layer and the channel width of meta-atoms which is defined by $\Gamma = \delta_v/w$.

[R4] G. Ward, R. Lovelock, A. Murray, A. P. Hibbins, J. R. Sambles, and J. Smith, Phys. Rev. Lett. 115, 044302 (2015).

[R5] X. Jiang, Y. Li, and L. Zhang, J. Acoust. Soc. Am. 141, EL363 (2017).

[R6] P. M. Morse and K. U. Ingard, Theoretical acoustics (Princeton university press, 1986).

[R7] T. Yazaki, Y. Tashiro, and T. Biwa, Proc. R. Soc. A: Math. Phys. Eng. Sci. 463, 2855 (2007).

Comment 3:

Although the focal spot implemented by the authors is subwavelength, the size of the entire metalens is many times that of the wavelength, and it can only work in waveguide environments, lacking engineering application value.

Reply 3:

We thank the reviewer for this comment. However, we respectfully argue on this aspect. A lens plays a pivotal role in manipulating the wavefront to achieve wave focusing. The lateral size of a lens is closely associated with accommodating radiant (luminous) energy to achieve high intensity and efficiency. In this context, in line with the fundamental lens mechanism, the lateral dimension size should be sufficient.

The notion of a thin form factor is a concept commonly employed not only in acoustic metasurfaces but also in

optical metasurfaces. The characteristic of metalenses lies in reducing the thickness at the subwavelength scale along the propagation axis (not the lateral axis), which is its original meaning. In most cases, across various lenses (including convex/concave lenses and metalenses), it is obvious that their lateral sizes—naturally spanning several wavelengths—are significant. Considering the *lateral size* of a metalens (not its *thickness*) at a *subwavelength scale* might appear contradictory. For instance, in conventional optical lenses used in cameras, their sizes (diameter) commonly range within tens to hundreds of wavelengths (at the scale of millimeters). Similarly, in acoustics, the size of the convex or concave acoustic lenses attached to ultrasonic transducers typically spans several to tens of wavelengths (at the scale of centimeters), and its thickness is typically a few wavelengths due to its geometric shape. For example, if we consider a subwavelength lateral size of metalens $D = 0.5 \lambda$ with the same focal length in our work $f = 2 \lambda$, the resulting NA is 0.12. Such *smaller-sized* lenses or those with lower NA values indeed have a limited capability to receive sufficient input waves, without doubt leading to diminished focusing intensity and a lower signal-to-noise ratio (SNR) from a sensing perspective.

In our work, the lateral size of the WFOH metalens is 8.4λ , designed to achieve a high NA (0.9 in this study). There are two aspects to consider in achieving a high NA. Firstly, a high NA signifies the capacity to gather ‘more energy’, resulting in improved focusing efficiency and enhanced SNR. Secondly, according to the definition of NA, $\sin[\arctan(\frac{D}{2f})]$, achieving a high NA means a relatively ‘shorter’ focal length. A shorter focal length implies a ‘smaller’ aperture-to-hearing plane distance, and a ‘minimized’ reception area $[-f, f]$. This implies the potential for creating a compact and robust sound reception system. These points have been further emphasized in the main text.

Furthermore, we understand the reviewer’s concern regarding the lack of engineering application value. We wish to address the concern about its limited functionality in *waveguide environments* by presenting evidence based on the numerical simulations. Fig. R2 demonstrates the 3D focusing fields, showcasing the wide-angle focusing capability in free space using a 2D metalens structure.

Fig. R2. Schematic of a 2D WFOH metalens in free space (not a waveguide environment) and calculated focusing fields by varying AOIs.

In principle, according to the section “*Details on symmetry conversion towards WFOH metalenses*” in Methods, the restriction of the waveguide environment is not necessary. However, we understand the reviewer’s concern regarding our realization of a 1D metalens. The objective of conducting experiments in a waveguide environment is to ensure an anechoic, noise-free experimental configuration for the 1D metalens, effectively eliminating unexpected nuisances to facilitate experimental verification. The underlying physics remains the same between 1D and 2D metalenses, and similar experimental results can be obtained if an anechoic chamber (free space environment) is prepared.

Change 1:

1. Please refer to lines 138-140 on page 4 of the revised manuscript.

“The use of such a high NA has two benefits: improved energy gathering tied to the capacity for radiant energy, and a shorter focal length (aperture-to-hearing plane distance), leading to a more compact and robust sound reception system.”

2. Please refer to lines 203-204 on page 4 of the revised manuscript.

“Its high NA and relatively short focal length ensure a small form factor (short distance of the aperture-to-hearing plane and minimized reception area $[-f, f]$) while maintaining focusing capability across a wide range of AOIs.”

Comment 4:

The implementation of related concepts in airborne has a greatly reduced significance, while it has a greater significance in ultrasound.

Reply 4:

We understand the reviewer’s concern on this matter. Intriguing applications in airborne scenarios related to our present work include source-tracking sound sensing, which can be extended to sound source localization. Additionally, utilizing the same mechanism, the next-generation acoustic camera is also promising. In this study, while our WFOH metalens is not limited to a specific frequency range, we determine the operation frequency in audible range to conduct well-developed experimental demonstrations, considering the constraints related to sample fabrication and measurement setup. In particular, the frequency range for measurements is limited to a maximum of 20 kHz due to the dynamic range of the loudspeaker array in our experimental set-up. Moreover, due to the resolution constraints imposed by 3D printing technology, there exists a lower bound for the sample size to ensure high-quality fabrication of the metalens structure with complex-shaped meta-atom geometry.

On the other hand, we agree with the reviewer's statement that the WFOH metalens holds greater significance within the ultrasonic frequency regime (or underwater applications if applied to audible frequencies). For example, our system has potential for high-impact applications such as non-destructive evaluation (NDE), medical treatment, and brain tumor imaging in the field of biology-related engineering, although the issue of impedance mismatch should be resolved [R8, R9]. But currently, it lies beyond the scope of the present work. We believe the efforts to showcase the applicability of ultrasound focusing, particularly concerning 3D structures, warrant a separate publication, leaving it for the next phase of our research along with the pursuit of achieving achromatic capability. However, we hold a strong conviction that our inaugural experimental showcase of the WFOH not only stands as a noteworthy proof but also provides a blueprint to propel future breakthroughs in acoustic metalens technology. We hope the reviewer will take this aspect into consideration.

[R8] L. Li, Y. Diao, H. Wu, and W. Jiang, *ACS Appl. Mater. Interfaces* 14, 28604 (2022).

[R9] S. Jimenez-Gambin, N. Jimenez, and F. Camarena, *Phys. Rev. Appl.* 14, 054070 (2020).

Response to Reviewer #2's comments:

Comment: This paper presents a design of a wide field of view acoustic metasurface, which can perform efficient acoustic focusing for a wide range of incident angles. The results appear quite interesting, however the presentation and explanation of the work is unclear, and in places appears contradictory.

Reply:

We sincerely thank the reviewer for the time and valuable comments, which have helped to greatly improve our manuscript. We are also delighted to note that the reviewer finds our work interesting. The authors have refined sentences throughout the manuscript and supplementary information, aiming to improve clarity and coherence. Moreover, a thorough proofreading was conducted to ensure the absence of any errata. We hope that the revised manuscript matches the high standard and diverse readership of *Nature Communications*. Please find below our detailed responses to the reviewer's comments.

Comment 1:

The authors start by noting that most metasurface lens designs consider the hyperbolic phase distribution. They then show how the ideal phase profile, which depends on the incident angle, has a relatively complex mathematical expression, and is not easy to implement in a realistic metamaterial structure. For this reason they introduce the quadratic phase profile (previously demonstrated for optical metasurfaces), which has a simpler dependence on the incident field.

To back up this claim that the quadratic phase profile is simpler, the authors refer to supplementary figure S1. However, it's not at all clear why Figure S1(b) should be considered a "flattened and undistorted phase distribution" when visually it appears much more complex than Figure S1(a) for the ideal phase profile. Furthermore, the quadratic phase profile as presented in Equation (1) appears to be angle dependent - yet in Figure S4, the authors demonstrate that (to a good approximation) their meta-atoms are angle-independent. I am able to clarify my understand by referring to the cited papers on optical wide field of view metalenses. These seem to indicate that the authors here are confusing the designed phase profile of the lens with the phase of the transmitted light in Eq. (1). So this confusion needs to be cleared up, and the manuscript made sufficiently self-contained that its key points can be understood, including by acoustics practitioners who may not be familiar with the work in optics.

Reply 1:

We would like to thank the reviewer for giving us an opportunity to clarify this very important point. In the original supplementary information, we presented the phase distribution of the quadratic metalens in Fig. S1(b), which might not be sufficiently straightforward. Indeed, as pointed out by the reviewer, Eq. (1) in the original manuscript (Eq. (2) in the revised manuscript) indicates the *effective* phase distribution of the *transmitted*

acoustic waves under *oblique illumination*. We would like to emphasize that this phenomenon is attributed to the symmetry conversion mechanism rather than the necessity for angle-dispersive meta-atom characteristics. Accordingly, the original Eq. (1) should be distinguished from the design phase: $\phi = -k_0 \frac{y^2}{2f}$. To avoid any potential confusion, we have revised Eq. (1) and Eq. (2) (please refer to “Change 1”) and have additionally included a detailed description of the quadratic phase and its symmetry conversion mechanism in both the main text and supplementary information. To provide additional details, particularly in Supplementary Note 2 which has been newly incorporated into the revised supplementary information, we offer a clear and comprehensive description.

It is important to note that symmetry conversion induces the *effective* phase shift $\phi_{PS} = -k_0 y \sin \theta_i$ under oblique illumination. Based on this, the *passively* designed metalens with the design phase profile $\phi = -k_0 \frac{y^2}{2f}$ can achieve a broad angular response. To provide an intuitive illustration of the effect of symmetry conversion, we have added Fig. S2, representing the actual phase distributions of the WFOH and conventional metalenses at exit pupils (the phase carried by the transmitted sound waves). To the best of our knowledge, exploring the actual phase distributions to explain symmetry conversion has not been undertaken previously in the optical domain. We believe that explanations based on the actual phase distribution offer a clearer understanding of symmetry transformation, especially for acoustic practitioners.

In Fig. S2(a), the effective translational shift of the phase distributions is evident despite the passive configuration of the designed metalens. While there are slight discrepancies between the theoretical predictions and both the FEM and experimental results, we still witness overall similar trends. On the other hand, in the case of conventional metalenses (Fig S2(b)), the desired phase distributions fail to form beyond 10 degrees, resulting in large aberrations that hinder wide-angle focusing.

Fig. S2: Phase distributions at the exit pupils of (a) WFOH and (b) conventional metalens; theoretical-solid lines, FEM-blue dots, and measured-red dots. Shaded area in (a) indicates the effective aperture (D_{eff}); outside this region, the wavevector becomes evanescent.

Change 1:

1. Please refer to lines 95-114 on page 4 of the revised manuscript.

“The WFOH system exploits the quadratic form of the one-dimensional phase profile, which is expressed as follows:

$$\phi(y) = -\frac{k_0}{2f}y^2, \quad (1)$$

where y represents the transverse position across the lens. The primary concept of Eq. (1) involves the transformation of a spherically symmetric gradient-index (Luneberg) lens in to a *flat-configuration* where both the lens’s geometrical shape and the focal plane can be flattened—achieving a perfect conversion from rotational symmetry to translational symmetry (Fig. 1(a)). If incident sound waves are projected in the xy -plane at an arbitrary AOI, the effective phase shift $\phi_{\text{PS}} = -k_0 y \sin \theta_i$ is incorporated, yielding the following phase distribution after transmitting through the metalens or at the exit pupil (EP) of the metalens:

$$\phi_{\text{EP}}(y, \theta_i) = -\frac{k_0}{2f}y^2 + \phi_{\text{PS}} = -\frac{k_0}{2f}(y + f \sin \theta_i)^2 + \frac{fk_0 \sin^2 \theta_i}{2}. \quad (2)$$

The effect of oblique incidence is evident in the transverse translation of the focal position by $f \sin \theta_i$, in Eq. (2). In the case of spherically symmetric lenses, the focal plane takes a non-planar form, requiring complex-receiving components; otherwise, it leads to field curvature aberration based on the planar focal plane. However, through a perfected symmetry conversion process, WFOH metalenses possess a planar receiving plane with predictable focal positions, enabling straightforward wide-angle acoustic sensing (Fig. 1(a)). Details on the derivation of the quadratic phase profile are provided in Methods. We emphasize that Eq. (2) does not imply the necessity for meta-atoms to possess angular dispersion, i.e., angle-dependent phase delay and transmission. Rather, it denotes the effective transformation from the self-contained phase shift under oblique illumination scenarios (see Supplementary Note 2 for a detailed discussion). Moreover, these characteristics enable successful realization comparable to a Fourier lens, applicable to diverse uses such as spatial filtering and compressed sensing⁶², all within a compact planar form factor.”

2. Please refer to page S3-S4 of the supplementary information (Supplementary Note 2, which is newly added).

“Supplementary Note 2: Wide-angle responses of WFOH metalenses by symmetry conversion

In this section, we delve into more details regarding the symmetry conversion based on the actual phase distributions at the exit pupils of metalenses.

As discussed in the main text, the WFOH metalens is designed with Eq. (1). The ability to achieve wide-angle focusing with angle-dispersion-free meta-atoms is attributed to the inherent characteristics of the quadratic phase. Specifically, it is owing to the effective phase shift ($\phi_{\text{PS}} = -k_0 y \sin \theta_i$) and its symmetry conversion property, which is induced by the oblique illumination. For direct investigation about this, we show the actual phase distributions at the exit pupils of metalenses, i.e., phases of the transmitted waves (Fig. S2).

Fig. S2(a) represents the case of the WFOH metalens, where the blue and red markers denote the results of FEM and experiment, respectively. Also, the shaded area indicates the effective aperture area, $[-f + f \sin \theta_i, f + f \sin \theta_i]$ (further discussed below). We can clearly observe an effective transverse shift in the phase distribution with different AOIs, despite the passive configuration of the designed metalens (see also the angle-invariant characteristics of meta-atoms, as shown in Fig. S5). Despite slight discrepancies from the theory in both FEM and experimental results, we can see overall similar trends. We note that various factors may contribute to deviations in the phase distribution of a real metalens from its ideal counterpart [S3], e.g., non-uniform transmittance of

meta-atoms, and non-local effects [S4, S5] between adjacent unit cells. Fig. S2(b) represents the phase distributions of metalenses designed based on the hyperbolic phase profile. It is apparent that beyond 10 degrees, the desired phase distributions fail to form, resulting in the inability to generate the ideal wavefront and consequently leading to aberrations that hinder wide-angle focusing.

Additionally, we characterize the symmetric properties of quadratic phase profile with the generalized law of refraction, which can be expressed as follows:

$$n_t \sin \theta_t - n_i \sin \theta_i = \frac{\lambda}{2\pi} \frac{d\phi}{dy} = \frac{k_y}{k_0}. \quad (\text{S2})$$

Here, n_t (n_i) indicate the refractive indices where the sound is transmitted (incident). We only consider one background medium (air) i.e., $n_t = n_i = 1$. From Eq. (S2), we see the normalized transverse wavenumber $\frac{k_y}{k_0}$ is related to the sound refraction with metasurface-induced phase discontinuity. The normalized transverse wavenumber of the quadratic phase can be expressed by

$$\frac{k_y}{k_0} = -\left(\frac{y}{f} - \sin \theta_i\right). \quad (\text{S3})$$

The quadratic phase maintains spatial symmetry along the axis of symmetry $f \sin \theta_i$, as shown in Eq. (S3). Based on Eq. (S3) and the condition $|k_y| > k_0$ for an imaginary wavevector along the acoustic axis $\left(k_x = \sqrt{k_0^2 - k_y^2}\right)$, we obtain the effective aperture as follows:

$$\begin{aligned} \frac{|k_y|}{k_0} = \left| \frac{y - f \sin \theta_i}{f} \right| > 1, \\ -f + f \sin \theta_i < y < f + f \sin \theta_i. \end{aligned} \quad (\text{S4})$$

Therefore, the effective aperture ($D_{\text{eff}} = 2f$) varies with $f \sin \theta_i$ in accordance with AOIs.”

3. Please refer to page S2 of the supplementary information (Supplementary Note 1)

“Supplementary Note 1: Phase surface map

We recall the angle-dependent (ideal) hyperbolic phase profile described in the main text,

$$\phi_{\text{ideal}}(r, \theta_i) = -k_0 \left(y \sin \theta_i + \sqrt{(y - y_0(\theta_i))^2 + f^2} - \sqrt{y(\theta_i)^2 + f^2} \right), \quad (\text{S1})$$

Where $y(\theta_i)$ is equal to $f \tan \theta_i$ which is the focal-point position along the focal plane. We

calculate the phase surface map with an NA of 0.9, as illustrated in Fig. S1(a). We note that this phase profile forms an undistorted (perfect) image on the imaging plane, as it is an ideal case determined by the focus offset $f \tan \theta_i$ [S1]. However, the angle-dependent nature poses a challenge in achieving an instantaneous real-time change of phase distribution with respect to θ_i once a passive design is implemented [S1, S2]. In this context, it is highly desirable to have a phase distribution along the surface of the metalens that is independent of variations in θ_i . One way to relax the angle-dependence is by using a small-angle approximation that yields $y_0(\theta_i) = f\theta_i$ (referred to as F-theta scan lenses in optics), as calculated in Fig. S1(b). This results in a relatively flattened phase distribution across the angle of incidence, though it comes with a distorted image mainly due to the change in $y_0(\theta_i)$. Still, in both cases, there are nonsymmetric phase distributions based on $\theta_i = 0$, making it challenging to anticipate consistent PSF formation with varying angles of incidence. In the following note, we further provide more details about the wide-angle response of the WFOH metalens.”

[S1] X. Luo, F. Zhang, M. Pu, Y. Guo, X. Li, and X. Ma, *Nanophotonics* 11, 1 (2021).

[S2] A. Kalvach and Z. Szabó, *JOSA B* 33, A66 (2016).

[S3] M. Zhao, M. K. Chen, Z.-P. Zhuang, Y. Zhang, A. Chen, Q. Chen, W. Liu, J. Wang, Z.-M. Chen, B. Wang, et al., *Light Sci. Appl.* 10, 52 (2021).

[S4] X. Wang, R. Dong, Y. Li, and Y. Jing, *Rep. Prog. Phys.* 86, 116501 (2023).

[S5] K. Shastri and F. Monticone, *Nat. Photonics* 17, 36 (2023).

Comment 2:

Figure 1 is intended to present the key ideas to the readers, but it does not do this job well. Subfigure (a) seems to contain nothing useful that is not shown more clearly in subfigure (b). Subfigure (c) shows the Fourier components of the point-spread function - it's unclear what useful information this conveys compared to just showing the point spread function. Finally, subfigure (d) appears to be just a plot of the well-known relationship between focal length, aperture size and numerical aperture, which does not seem to warrant an entire figure.

Reply 2:

We would like to thank the reviewer for pointing this out. Taking the valuable feedback into account, we have made significant modifications to Fig. 1 in order to convey the main concept of our study properly. In Fig. 1(a), we have included a modified schematic illustration depicting the symmetry conversion process, representing how a flat WFOH metalens functions equivalently to a spherical symmetric lens. Additionally, as previously described in “Reply 1”, we provide a clearer explanation of the physical principles underlying the WFOH metalens in the revised manuscript and supplementary information.

Moreover, we would like to retain the MTF graph (Fig. 1(b)) but change the label from ‘Contrast’ to ‘MTF’ (our original intention in using ‘Contrast’ was to avoid jargon that might be unfamiliar notion to acoustic practitioners). To provide a more comprehensive description of the MTF, we have refined the main text to convey its physical significance concerning spatial frequency components and resolution, focusing on the imaging (or hearing) perspective. In addition, we have included 2D focusing field plots in Fig. 1(c) to directly compare angular responses, aiming to visually clarify the distinctions between WFOH and conventional metalenses. We anticipate that potential readers will easily grasp the main idea of this work with the revised Figure 1.

Fig. 1 Principle of a wide FOH system. (a) Schematic illustration of the acoustic symmetry conversion that transforms rotational symmetry under oblique incidence into transitional symmetry on the receiving plane with the transverse focal shift. In this process, both the spherically symmetric lens shape with a non-planar receiving plane is effectively converted to a thin-flat lens with a planar receiving plane. FOH refers to the hearable angular range, analogous to FOV in the vision system, determined by the reception angle limit, $\pm\theta_{\max}$. (b) MTF comparison between WFOH and conventional lenses with an NA of 0.9 as a function of AOI. Due to substantial off-axis aberrations, the MTF of the conventional case drastically decreases with increasing AOIs, leading to a loss in focusing ability. Conversely, the WFOH case demonstrates consistent and flattened MTFs regardless of AOIs and spatial frequency components, indicating its wide-angle focusing capability. (c) Theoretical prediction of the focusing field for ideal 2D lenses: WFOH metalenses with small aberrations (left) and conventional metalenses with large aberrations (right) under varying AOIs.

Change 1:

1. Please refer to lines 122-126 on page 4 of the revised manuscript.

“... MTF typically decreases for higher spatial frequencies, based on $2 \times \text{NA} / \lambda$ (diffraction limit), indicating that the NA imposes limitations on spatial resolution. Specifically, high f_y corresponds

to finer details within an object, such as edge information. Conversely, low f_y relates to the broader contours and general features of the object. Therefore, the analysis of MTF and PSF is crucial for characterizing a lens's performance, particularly across various incident angles.”

2. Please refer to lines 133-154 on page 5 of the revised manuscript.

“..., In Fig. 1(c), we can observe that the PSFs of conventional metalenses exhibit large aberrations at large AOIs, accompanied by low focusing efficiency and unpredictable focal positions. In contrast, WFOH metalenses have robust PSFs regardless of AOIs which indicate wide-angle focusing capability.

Without loss of generality, we now target a high NA of approximately 0.9 with fixed values of f and D . The use of such a high NA has two benefits: improved energy gathering tied to the capacity for radiant energy, and a shorter focal length (aperture-to-hearing plane distance), leading to a more compact and robust sound reception system. ..., The wavevector of transmitted acoustic waves decomposes into the wavevector component along the acoustic axis $k_x = \sqrt{k_0^2 - k_y^2}$, and $k_y = \frac{d\phi}{dy}$ along the transverse direction induced by the phase gradient. When $|k_y| > k_0$, k_x becomes imaginary, thereby converting the sound waves into evanescent waves through total internal reflection, which does not contribute to the focusing. ..., Interestingly, the underlying physics of the evanescent zone involves the formation of a self-adjustable aperture stop. This feature effectively suppresses additional marginal rays that might otherwise contribute to the onset of coma aberrations.”

Comment 3:

The authors mention several times the "translational symmetry" of their design. This is not explained well, but would make sense once Eq (1) is properly explained.

Reply 3:

We agree with the reviewer's comment that the explanation of "translational symmetry" was not clear in the original manuscript. As discussed in "Reply 1" and "Reply 2", we have provided a detailed explanation of the symmetry conversion mechanism of the quadratic phase in the revised manuscript and supplementary information. In particular, the mathematical description of the quadratic phase and actual phase distributions induced by its symmetry conversion effect are included in Methods and Supplementary Note 2.

Comment 4:

In the supplementary information, the authors show that their meander line structures can only achieve full transmission when their transmission phase is 0 - due to the use of Fabry-Perot resonances. For this reason additional Helmholtz resonators are introduced - it's unclear whether the meander lines are really necessary, or if the Helmholtz resonators would be sufficient on their own (as they have been shown in other designs - although typically 3 or more Helmholtz resonators are required in such cases).

Reply 4:

We thank the reviewer for pointing out the design perspective. Indeed, the meander line structure (zigzag channel) has limitations in achieving both full transmission and 2π phase modulation simultaneously, as discussed in Supplementary Note 3. With the design of Helmholtz resonators in series raised by the reviewer, it is possible to obtain near-unitary transmission by matching acoustic impedance with a half-wavelength waveguide and achieving full phase coverage with Helmholtz resonators as purely imaginary impedances (reactance). However, achieving independent control of transmission and phase in such a design is quite challenging. In this manner, we tried to make efforts to have a more generic design recipe by constructing the hybrid configuration composed of Helmholtz resonators and a zigzag channel. Although our current work is dedicated to achieving full (not arbitrary) transmission and arbitrary phase, our proposed meta-atom is flexible in terms of independent control of transmission and phase. This versatility can be further applied to contact-free particle manipulation (acoustic tweezer), or acoustic holography, and non-local metasurfaces. We note that the WFOH metalens is not highly limited by our hybrid meta-atom configuration though. This response also applies to “Comment 6” and “Reply 6”.

Comment 5:

Note that the claim on page 4 that the Helmholtz resonators "amplify the signal" is misleading - since these are passive structures. Possibly the authors are referring to impedance matching with the Helmholtz resonators.

Reply 5:

Thank you for the technical comment. We entirely agree with the claim raised by the reviewer. We have deleted the statement “amplify the signal” as it might be misleading, given that it is not a gain medium. According to the suggestion, we have revised the manuscript accordingly.

Change 5:

Please refer to lines 159 on page 5 of the revised manuscript.

“..., we integrate additional HRs for impedance matching and provide phase compensation.”

Comment 6:

On page 5 the authors mention "simultaneous modulation of T and Φ " - yet it seems that they only require structures with (approximately) unity transmission, so only modulation of Φ is relevant to this work.

Reply 6:

We agree with the reviewer's comment that only the phase modulation is relevant to our work. We recognize the potential for misunderstanding in our original statement and have consequently changed the sentence. Please find a more detailed response in "Reply 4".

Change 6:

1. Please refer to lines 159-160 on page 5 of the revised manuscript.

"..., we can obtain the simultaneous modulation of T and ϕ ⁶⁴⁻⁶⁶ by adjusting geometric parameters w_4 and h_1 ;"

2. Please refer to lines 162-165 on page 5 of the revised manuscript.

"We note that our meta-atoms can also find utility in applications such as acoustic holography and particle manipulation, even though we only consider the phase modulation in this work to achieve highly efficient focusing."

[64] Tian, Y., Wei, Q., Cheng, Y. & Liu, X. Acoustic holography based on composite metasurface with decoupled modulation of phase and amplitude. *App. Phys. Lett* 110, 191901 (2017).

[65] Ghaffarivardavagh, R., Nikolajczyk, J., Glynn Holt, R., Anderson, S. & Zhang, X. Horn-like space-coiling metamaterials toward simultaneous phase and amplitude modulation. *Nat. Commun.* 9, 1349 (2018).

[66] Zhu, Y. et al. Fine manipulation of sound via lossy metamaterials with independent and arbitrary reflection amplitude and phase. *Nat. Commun.* 9, 1632 (2018).

Comment 7:

The Experimental realization section is the strongest part of this paper - the results clearly show good performance, the agreement between experiments and numerics is fairly good, and the discussion/explanation is easy to follow.

Reply 7:

We are grateful to the reviewer for his/her positive assessment of the most crucial section, '*Experimental realization of wide field-of-hearing metalens*', as it not only marks the first realization but also serves as a new benchmark with demonstrations.

Comment 8:

In the abstract and discussion section, it is claimed that this is an "ultrathin and highly compact design" - but based on page 5 it is stated that $w_0=0.75*\lambda$, which is comparable to the wavelength. On the other hand, $h_0=\lambda/12.5$ is relatively small - this is essential for achieving the wide field of hearing.

Reply 8:

We appreciate the reviewer's comment. Although we have avoided using the term "ultrathin", it is worth noting that w_0 is still smaller than the wavelength, leading us to use the term "thin" instead. Indeed, h_0 is relatively small, satisfying the spatial version of Nyquist-sampling theorem described in the main text. This implies that discretized meta-atoms corresponding to the encoded phase behave like a continuous (ideal) phase, minimizing potential loss and aberrations from spurious (or parasitic) diffraction caused by aliasing.

Comment 9:

To understand the significance of the achieved results, it is necessary for the reader to look at Supplementary Figure S7. I think it would be much clearer if sub-figure S7(e) was somehow incorporated into the main text.

Reply 9:

We thank the reviewer for the valuable suggestion. As suggested by the referee, we have included Fig. S7(e) from the supplementary information as a subfigure in Fig. 4 to enhance the significance of our work.

Fig. 4. Demonstration of wide-angle sound-reception system. (a) Schematic of sound monitoring with conventional and WFOH metalenses. (b) Sound intensity distributions of the WFOH (left panel) and conventional metalens (right) on the receiving focal plane, ranging from 0° to 60° in AOIs; measured data-dots (only for the left panel), theoretical results-solid lines and FEM results-dotted lines. In the left panel, dashed gray lines labeled 'Threshold' represent the cosine-dependent relative illumination effect. Vertical bars in the right panel indicate the desired focal spot of the conventional metalens, $f \tan \theta_i$. (c) Transverse focal-spot location of the WFOH metalens as a function of AOI. The measured points are shown by dots, and the solid line indicates the theoretical $f \sin \theta_i$ curve. (d) Calculated focusing efficiencies of WFOH and conventional metalenses as a function of AOI.

Change 9:

1. Please refer to lines 206-214 on page 8 and 9 of the revised manuscript.

“In the left panel of Fig. 4(b), PSFs of the WFOH metalens obtained by normalizing line intensity on the focal plane exhibit excellent agreement between the theoretical, FEM, and experimental results, emphasizing the accuracy and reliability of the proposed design. On the other hand,

as shown in the right panel of Fig. 4(b), the PSFs indicate that the conventional metalens only works effectively at normal incidence. Under oblique incidence, the actual focal spots deviate significantly from the desired focal spot denoted by pink vertical bars following $f \tan \theta_i$, accompanied by considerable intensity suppression due to large off-axis aberrations.”

Comment 10:

In Figure 1 caption and on page 4 of the main text, the term "perfect conversion from rotational symmetry to translational symmetry" is used, without proper explanation. Either this concept should be explained properly in the main text, or just removed entirely.

Reply 10:

We agree with the reviewer’s comment that the explanation of ‘perfect conversion from rotational symmetry to translational symmetry’ was insufficient in the original manuscript. In response, as described in “Reply 1” and “Reply 2”, we have extensively elaborated on the physical principle responsible for the symmetry conversion of the quadratic phase in the revised manuscript and supplementary information, complemented by the revised Fig. 1 and Fig. S2. We think that our revised version can be easily understood by acoustics practitioners who are not familiar with metrics for characterization of the lens’s performance, such as FOH, aberration, NA, etc.

Comment 11:

In Figure 1(c), it is confusing to use "contrast" for a quantity when the term "MTF" is used in the main text - apparently referring to the same thing.

Reply 11:

We thank the reviewer for this comment. We have replaced the term “contrast” with “MTF” to avoid any confusion. Additionally, as outlined in “Reply 2”, we have included a detailed explanation of modulation transfer function in the revised manuscript to make it clear for acoustic practitioners.

Comment 12:

In Equation (1), the term x should be removed, since only the 1D case is considered in all other equations.

Reply 12:

We thank the reviewer for this careful comment. We have corrected Eq. (1) accordingly.

Comment 13:

In Figure 4, the line labeled "Threshold" should be better explained in the caption (it is mentioned in the main text, but with a different name)

Reply 13:

We thank the reviewer for pointing out the missing point. We have added a description of cosine dependent illumination effect to the caption in Figure 4.

Change 13:

Please refer to Figure 4 on page 8 of the revised manuscript.

“(b) ..., dashed gray lines labeled ‘Threshold’ represent the cosine-dependent relative illumination effect.”

Comment 14:

Overall this paper shows interesting experimental results, but the accompanying theoretical analysis and explanation need to be made much clearer.

Reply 14:

We are excited about the positive evaluation of the experimental results by the reviewer and grateful for the invaluable insights shared by Reviewer #2 that significantly enhanced our manuscript. His/her remarks allowed us to present our experimental results more clearly, refine the theoretical analysis, and provide explanations regarding the physical principles of the WFOH system. We believe that the revised version now provides a clear understanding of the core of our research.

Response to Reviewer #3's comments:

Comment: In this study, the authors proposed an acoustic metalens exhibiting wide field-of-hearing (FOH), which is an acoustical analogy of the wide field-of-view (FOV) in optics. The acoustic metalens is composed of several unit structures with different phase shifts to satisfy the phase distribution for wide FOH. Each unit structure, constructed from Helmholtz resonators with zigzag channels, exhibited high transmission and various phase shifts from 0 to 2π . The phase shift and transmission of the unit structures were calculated using a theoretical model derived by the authors and the results were validated by FEM simulations. The characteristics of the wide FOH metalens also were verified by experiments. Despite aforementioned efforts, the reviewer is not convinced of significant contribution in this work for the following reasons.

Reply:

We express our sincere gratitude to Reviewer #3 for dedicating time and providing helpful feedback that greatly improved our manuscript. The authors have made changes to sentences in the manuscript and supplementary information with the intention of enhancing clarity and coherence. We have tried to incorporate all of them to a satisfactory level for the high standard and diverse readership of *Nature Communications*. Please find below our detailed responses to the reviewer's comments.

Comment 1:

The key to the wide FOH is the specific phase distribution discovered already in terms of FOV by Pu et al. [Opt. Express, 25, 31471–31477 (2017)]. In the manuscript, the phase distribution was employed without any modification or extension when bringing it from the optical to the acoustical domain.

Reply 1:

We appreciate the reviewer's comment. Fundamentally, the optical and acoustic domains share common physical principles, particularly concerning wavefront engineering. Hence, it is a fact that our current work employs the same phase distribution (i.e., quadratic phase) discovered by Pu et al., as noted by the reviewer.

We would like to emphasize that the first claim on the necessity of FOH, along with the experimental realization of unprecedented WFOH within the acoustic metalens community, represents the significant novelty. Moreover, by quantifying the lens's performance with considerations such as NA, aberrations, MTF, and focusing efficiency—measures often overlooked in previous studies—we aim to provide valuable insights and guide towards potential advancements in acoustic metalens technology beyond mere proof-of-concept demonstrations of field focusing measures. With our introduction of the wide FOH capability, we strongly believe our present work holds substantial contributions. We hope that the reviewer acknowledges the dedication we have put into this work and sees it as a significant contribution to the community.

To address and back up the physics aspects of concern raised by the referee, we have enriched the revised manuscript. This includes extensive explanations of the physical principles underlying the WFOH system within the main text. Additionally, we have specifically provided a detailed investigation of the actual phase distributions at exit pupils in Supplementary Note 2. As far as we know, exploring the actual phase distributions to explain symmetry conversion has not been previously undertaken in the optical domain. We firmly believe that our revised explanations provide a clearer understanding for acoustic practitioners, facilitating a more intuitive comprehension.

Change 1:

1. Please refer to lines 95-114 on page 4 of the revised manuscript.

“The WFOH system exploits the quadratic form of the one-dimensional phase profile, which is expressed as follows:

$$\phi(y) = -\frac{k_0}{2f}y^2, \quad (1)$$

where y represents the transverse position across the lens. The primary concept of Eq. (1) involves the transformation of a spherically symmetric gradient-index (Luneberg) lens in to a *flat-configuration* where both the lens’s geometrical shape and the focal plane can be flattened—achieving a perfect conversion from rotational symmetry to translational symmetry (Fig. 1(a)). If incident sound waves are projected in the xy -plane at an arbitrary AOI, the effective phase shift $\phi_{PS} = -k_0y \sin \theta_i$ is incorporated, yielding the following phase distribution after transmitting through the metalens or at the exit pupil (EP) of the metalens:

$$\phi_{EP}(y, \theta_i) = -\frac{k_0}{2f}y^2 + \phi_{PS} = -\frac{k_0}{2f}(y + f \sin \theta_i)^2 + \frac{fk_0 \sin^2 \theta_i}{2}. \quad (2)$$

The effect of oblique incidence is evident in the transverse translation of the focal position by $f \sin \theta_i$, in Eq. (2). In the case of spherically symmetric lenses, the focal plane takes a non-planar form, requiring complex-receiving components; otherwise, it leads to field curvature aberration based on the planar focal plane. However, through a perfected symmetry conversion process, WFOH metalenses possess a planar receiving plane with predictable focal positions, enabling straightforward wide-angle acoustic sensing (Fig. 1(a)). Details on the derivation of the quadratic phase profile are provided in Methods. We emphasize that Eq. (2) does not imply the necessity for meta-atoms to possess angular dispersion, i.e., angle-dependent phase delay and transmission. Rather, it denotes the effective transformation from the self-contained phase shift under oblique illumination scenarios (see Supplementary Note 2 for a detailed discussion). Moreover, these characteristics enable successful realization comparable to a Fourier lens, applicable to diverse

uses such as spatial filtering and compressed sensing⁶², all within a compact planar form factor.”

2. Please refer to page S3-S4 of the supplementary information (Supplementary Note 2, which is newly added).

“Supplementary Note 2: Wide-angle responses of WFOH metalenses by symmetry conversion

In this section, we delve into more details regarding the symmetry conversion based on the actual phase distributions at the exit pupils of metalenses.

As discussed in the main text, the WFOH metalens is designed with Eq. (1). The ability to achieve wide-angle focusing with angle-dispersion-free meta-atoms is attributed to the inherent characteristics of the quadratic phase. Specifically, it is owing to the effective phase shift ($\phi_{PS} = -k_0 y \sin \theta_i$) and its symmetry conversion property, which is induced by the oblique illumination. For direct investigation about this, we show the actual phase distributions at the exit pupils of metalenses, i.e., phases of the transmitted waves (Fig. S2).

Fig. S2(a) represents the case of the WFOH metalens, where the blue and red markers denote the results of FEM and experiment, respectively. Also, the shaded area indicates the effective aperture area, $[-f + f \sin \theta_i, f + f \sin \theta_i]$ (further discussed below). We can clearly observe an effective transverse shift in the phase distribution with different AOIs, despite the passive configuration of the designed metalens (see also the angle-invariant characteristics of meta-atoms, as shown in Fig. S5). Despite slight discrepancies from the theory in both FEM and experimental results, we can see overall similar trends. We note that various factors may contribute to deviations in the phase distribution of a real metalens from its ideal counterpart [S3], e.g., non-uniform transmittance of meta-atoms, and non-local effects [S4, S5] between adjacent unit cells. Fig. S2(b) represents the phase distributions of metalenses designed based on the hyperbolic phase profile. It is apparent that beyond 10 degrees, the desired phase distributions fail to form, resulting in the inability to generate the ideal wavefront and consequently leading to aberrations that hinder wide-angle focusing.

Additionally, we characterize the symmetric properties of quadratic phase profile with the generalized law of refraction, which can be expressed as follows:

$$n_t \sin \theta_t - n_i \sin \theta_i = \frac{\lambda}{2\pi} \frac{d\phi}{dy} = \frac{k_y}{k_0}. \quad (S2)$$

Here, n_t (n_i) indicate the refractive indices where the sound is transmitted (incident). We only consider one background medium (air) i.e., $n_t = n_i = 1$. From Eq. (S2), we see the normalized

transverse wavenumber $\frac{k_y}{k_0}$ is related to the sound refraction with metasurface-induced phase discontinuity. The normalized transverse wavenumber of the quadratic phase can be expressed by

$$\frac{k_y}{k_0} = -\left(\frac{y}{f} - \sin \theta_i\right). \quad (\text{S3})$$

The quadratic phase maintains spatial symmetry along the axis of symmetry $f \sin \theta_i$, as shown in Eq. (S3). Based on Eq. (S3) and the condition $|k_y| > k_0$ for an imaginary wavevector along the acoustic axis $\left(k_x = \sqrt{k_0^2 - k_y^2}\right)$, we obtain the effective aperture as follows:

$$\frac{|k_y|}{k_0} = \left|\frac{y - f \sin \theta_i}{f}\right| > 1, \\ -f + f \sin \theta_i < y < f + f \sin \theta_i. \quad (\text{S4})$$

Therefore, the effective aperture ($D_{\text{eff}} = 2f$) varies with $f \sin \theta_i$ in accordance with AOIs.”

Fig. S2: Phase distributions at the exit pupils of (a) WFOH and (b) conventional metalens; theoretical-solid lines, FEM-blue dots, and measured-red dots. Shaded area in (a) indicates the effective aperture (D_{eff}); outside this region, the wavevector becomes evanescent.

3. Please refer to lines 226-227 on page 9 of the revised manuscript.

“In summary, we introduced a new metric, FOH—previously overlooked in existing acoustic metalenses—and showcased the hallmark of WFOH capability.”

Comment 2:

Besides, discussions for the underlying physics of wide FOH in the supplementary information were almost the same with those in a recent review paper [Nanophotonics, 11(1), 1-20 (2022)]. It seems that ‘phase surface contours’ presented in the supplementary information were similar to those in the Nanophotonics paper with only the transformation of the x-axis from $\sin\theta$ to θ : nevertheless, this paper was not referred in both the manuscript and supplementary information.

Reply 2:

We apologize for this mistake. In response to the reviewer’s concern, we have properly incorporated two references in both the main text and supplementary information. The points we have raised are as follows: the realization of an ideal phase lens ensures an undistorted image with well-constructed PSFs along the focal plane, but only if the condition of angle-dispersive meta-atoms is required. Plus, although the small-angle approximation (usually termed the F-theta scan lens in conventional optics) can alleviate the angle-dependence to some extent, resulting in a relatively flattened phase surface map across AOIs; yet, the nonsymmetric phase distribution still contributes to off-axis aberrations. However, unlike the need for angle-dispersive characteristics in the ideal phase profile, the quadratic phase does not necessitate such features. Passively designed metalenses employing the quadratic phase profile (Eq. (1)) enable the realization of WFOH capability. It is important to note that symmetry conversion induces the *effective* phase shift $\phi_{PS} = -k_0 y \sin \theta_i$ under oblique illumination (refer to Eq. (2)). To intuitively offer the effect of symmetry conversion, we have added Fig. S2, illustrating the actual phase distributions of the WFOH and conventional metalenses at exit pupils (the phase carried by the transmitted sound waves). This additional data serves as a distinctive signature, providing strong support for the effectiveness of the quadratic phase in achieving wide-angle capability.

Change 2:

1. Please refer to lines 82-84 on page 3 of the revised manuscript.

“However, achieving an ideal phase profile heavily relies on nonsymmetric behavior with respect to the AOI in oblique illumination scenarios^{54,55}. This indicates the necessity for angle-dispersive properties that cannot be achieved with passive structures (see Supplementary Note 1 for details).”

[54] Kalvach, A. & Szabo, Z. Aberration-free flat lens design for a wide range of incident angles. *JOSA B* 33, A66–A71 (2016).

[55] Luo, X. et al. Recent advances of wide-angle metalenses: principle, design, and applications. *Nanophotonics* 11, 1–20 (2021).

2. Please refer to page S2 of the supplementary information (Supplementary Note 1)

“Supplementary Note 1: Phase surface map

We recall the angle-dependent (ideal) hyperbolic phase profile described in the main text,

$$\phi_{\text{ideal}}(r, \theta_i) = -k_0 \left(y \sin \theta_i + \sqrt{(y - y_0(\theta_i))^2 + f^2} - \sqrt{y(\theta_i)^2 + f^2} \right), \quad (\text{S1})$$

Where $y(\theta_i)$ is equal to $f \tan \theta_i$ which is the focal-point position along the focal plane. We calculate the phase surface map with an NA of 0.9, as illustrated in Fig. S1(a). We note that this phase profile forms an undistorted (perfect) image on the imaging plane, as it is an ideal case determined by the focus offset $f \tan \theta_i$ [S1]. However, the angle-dependent nature poses a challenge in achieving an instantaneous real-time change of phase distribution with respect to θ_i once a passive design is implemented [S1, S2]. In this context, it is highly desirable to have a phase distribution along the surface of the metalens that is independent of variations in θ_i . One way to relax the angle-dependence is using a small-angle approximation that yields $y_0(\theta_i) = f\theta_i$ (referred to as F-theta scan lenses in optics), as calculated in Fig. S1(b). This results in a relatively flattened phase distribution across the angle of incidence, though it comes with a distorted image mainly due to the change in $y_0(\theta_i)$. Still, in both cases, there are nonsymmetric phase distributions based on $\theta_i = 0$, making it challenging to anticipate consistent PSF formation with varying angles of incidence. In the following note, we further provide more details about the wide-angle response of the WFOH metalens.”

[S1] X. Luo, F. Zhang, M. Pu, Y. Guo, X. Li, and X. Ma, *Nanophotonics* 11, 1 (2021).

[S2] A. Kalvach and Z. Szabó, *JOSA B* 33, A66 (2016).

Comment 3:

Regarding the implementation of the wide FOH, acoustic metalenses in a paper [J. Phys. D, 52, 385303 (2019)] also showed that the wide angles of incidence could be allowed to 60 degrees, but the manuscript did not provide a concrete comparison with this or any other related works. As for the unit structure, the width of the unit structure in the manuscript was thinner than that in the J. Phys. D, but one in J. Phys. D also could be sufficiently

thin to satisfy the rapid phase gradient necessary for implementing a wide FOH metalens when considering the results for the wide angles of incidence as mentioned above.

Reply 3:

We respectfully disagree with the reviewer’s opinions about the paper [52, S16 - *J. Phys. D*, 52, 385303 (2019)]. We thoroughly reviewed the paper [52, S16] and hope that any misunderstandings will be resolved. The authors in that study employed a modified form of the hyperbolic phase profile as follows:

$$\phi(x) = -k_0 \left(\sqrt{x^2 + y^2 + f^2} + y \sin \theta_i \right) + \phi_0.$$

In the supplementary data of the mentioned paper, the authors specifically designed the metasurface referred to as AMS-x1-y1-60°. It was designed to steer the incoming acoustic waves at a 60-degree angle of incidence (AOI) and converge at the focal point along the acoustic axis perpendicular to the lens. In Fig. R3, we represent the phase profile of AMS-x1-y1-60°.

Fig. R3. Phase profile of AMS-x1-y1-60°.

We want to emphasize that this phase profile enables wave focusing specifically for 60-degree angle of incidence, rather than for a *broad range* of incident angles with a single device. Deviating from the single target angle (such as normal incidence in the hyperbolic phase), off-axis aberrations inevitably occur, primarily stemming from the inherent characteristics of the hyperbolic-phase-based metalens. Here, we calculate the acoustic intensity fields of AMS-x1-y1-60° based lens (please refer to Fig. R4). We consider a NA of 0.8, consistent with [52, S16]. We can observe a significant presence of off-axis aberrations in the calculated acoustic fields. Again, a lens designed with a phase profile like AMS-x1-y1-60° cannot achieve the wide-angle focusing capability due to its inherent nature as a hyperbolic-phase-based lens.

Fig. R4. Acoustic field distribution of AMS-x1-y1-60° at (a) 60, (b) 40, and (c) 20 AOIs, respectively.

In addition, we show one-dimensional intensity profiles of AMS-x1-y1-60° with different AOIs (Fig. R5). We observe significant aberrations and side lobes that hinder the resolution capability, evident in inconsistent MTFs across AOIs, low focusing efficiency, and non-matched desired focal spots (leading to field-curvature aberrations). It is clear that the aberrations occur when deviating more than 20-degree from the 60-degree reference angle, contradicting the claimed WFOH capability suggested by the reviewer.

Fig. R5. 1D intensity profiles of AMS-x1-y1-60° with different AOIs.

Moreover, in response to the reviewer's comment "... the manuscript did not provide a concrete comparison with this or any other related works ...", we have conducted a comprehensive literature survey on acoustic metalenses in Supplementary Table 1. This table includes [52, S16] and it has been added to the Supplementary Note 9.

[52, S16] K. Gong, X. Wang, H. Ouyang, and J. Mo, *J. Phys. D: Appl. Phys.* 52, 385303 (2019).

Change 3:

1. Please refer to page S15-S16 of the supplementary information (Supplementary Note 9, which is newly added).

“Supplementary Note 9: Comparison with previous studies on acoustic metalenses

We compare our work with previous studies on acoustic focusing using planar metasurfaces. A detailed comparison, as shown in Table S1, highlights crucial performance metrics of the lens, particularly for the FOH. To our knowledge, the determination of the size and focal length of metalenses seems arbitrary in Refs. [S10-S25], revealing a deficiency in considering NA during the design process, often due to the predominant focus on proof-of-concept demonstrations regarding focusing capabilities. By addressing the existing limitations of acoustic metalenses in achieving WFOH while adhering to the hyperbolic phase profile (often recognized as the trade-off between high-NA and wide FOV in conventional metalenses [S26]), our current work significantly enhances wide angle focusing capability—an aspect often overlooked in previous studies—by emphasizing the necessity of considering FOH. While Ref. [S12] presented results for oblique incidence, it lacked detailed interpretation with its narrow FOH.”

- [S10] C. Kim, J. Kim, and W. Jeon, *J. Sound Vib.* 529, 116910 (2022).
- [S11] W. Li, F. Meng, and X. Huang, *Appl. Phys. Lett.* 117 (2020).
- [S12] P. Wang, G. Yu, Y. Li, X. Wang, and N. Wang, *New J. Phys.* 22, 023006 (2020).
- [S13] F. Zhang, E. Perkins, S. Wang, G. T. Flowers, and R. N. Dean, *Appl. Phys. Express* 12, 087002 (2019).
- [S14] J. Chen, J. Xiao, D. Lisevych, A. Shakouri, and Z. Fan, *Nat. Commun.* 9, 4920 (2018).
- [S15] S. Tang, B. Ren, Y. Feng, J. Song, and Y. Jiang, *J. Appl. Phys.* 129 (2021).
- [S16] K. Gong, X. Wang, H. Ouyang, and J. Mo, *J. Phys. D: Appl. Phys.* 52, 385303 (2019).
- [S17] S. Qi and B. Assouar, *J. Appl. Phys.* 123 (2018).
- [S18] N.-L. Zhang, S.-D. Zhao, H.-W. Dong, Y.-S. Wang, and C. Zhang, *Appl. Phys. Lett.* 120 (2022).
- [S19] X. Jiang, Y. Li, D. Ta, and W. Wang, *Phys. Rev. B* 102, 064308 (2020).
- [S20] L. Xiang, L. Jian, and H. Xinjing, *IEEE Sens. J.* 22, 13989 (2022).
- [S21] Y.-F. Zhu, X.-D. Fan, B. Liang, J. Yang, J. Yang, L.-l. Yin, and J.-C. Cheng, *AIP Adv.* 6 (2016).
- [S22] Y. Zhu and B. Assouar, *Phys. Rev. B* 99, 174109 (2019).
- [S23] H.-W. Dong, C. Shen, S.-D. Zhao, W. Qiu, H. Zheng, C. Zhang, S. A. Cummer, Y.-S. Wang, D. Fang, and L. Cheng, *Natl. Sci. Rev.* 9, nwac030 (2022).
- [S24] J.-K. Weng, Y.-F. Zhu, B. Liang, J. Yang, and J.-C. Cheng, *Appl. Phys. Express* 13, 094003 (2020).
- [S25] S.-D. Zhao, A.-L. Chen, Y.-S. Wang, and C. Zhang, *Phys. Rev. Appl.* 10, 054066 (2018).

Table S1. Summary of acoustic metalenses.

Ref.	NA	FOH (deg)	Type	Exp. verification	Thickness (λ)	FWHM (λ)	Focusing efficiency (%)	Operation frequency (kHz)
This work	0.9	140	2D/ Transmissive	O	0.75	0.39-0.42	41-43	3.43
[S10]	0.76	N/A	2D/ Transmissive	O	0.5	0.56	N/A	2.5, 4, 5.5
[S11]	0.74	N/A	2D/ Transmissive	O	0.54	0.52	41.5	3.1
[S12]	0.51	20	2D/ Reflective	O	0.5	N/A	N/A	2.8-5.6
[S13]	0.49	N/A	2D/ Transmissive	O	0.87	N/A	N/A	7.5
[S14]	0.5	N/A	3D/ Transmissive	O	0.05	0.98	N/A	3.43
[S15]	0.68	N/A	2D/ Transmissive	O	0.5	N/A	N/A	2.7, 4.1, 5.5
[S16]	0.8, 0.94	N/A	2D/ Transmissive	X	0.65	0.52, 0.39	N/A	5
[S17]	0.68, 0.73	N/A	3D/ Reflective	X	0.07	0.76, 0.74	N/A	3.43
[S18]	0.7	N/A	2D/ Reflective	O	0.2	0.64-0.75	N/A	3, 3.5, 4.5
[S19]	0.15	N/A	2D/ Reflective	X	0.25	0.62	N/A	500 (water)
[S20]	0.87	N/A	3D/ Transmissive	O	0.98	0.55-0.83	N/A	8
[S21]	0.86	N/A	2D/ Reflective	X	0.5	N/A	N/A	1.7, 3.4, 5.1
[S22]	0.78	N/A	2D/ Reflective	O	0.21	N/A	N/A	2
[S23]	0.79	N/A	2D/ Transmissive	O	0.87	0.59-0.7	N/A	1-4
[S24]	0.68	N/A	2D/ Reflective	X	0.25	N/A	N/A	1.3, 1.9, 2.4
[S25]	0.88	N/A	3D/ Transmissive	O	0.8	\sim 0.6	N/A	5.5

Comment 4:

Based on the concerns raised regarding novelty and significant improvement over prior works, the reviewer cannot recommend the publication of the present manuscript in Nature Communications.

Reply 4:

We would like to express our gratitude once again to Reviewer #3 for her/his valuable feedback. We have made efforts to incorporate all the suggestions to a satisfactory level. We hope the reviewer finds the revised manuscript satisfactory and eagerly anticipates receiving positive feedback in a second round.

Reviewer #1 (Remarks to the Author):

Although the authors have made great efforts to improve the quality of the manuscript, I still believe that it cannot meet the requirements of NC from the novelty perspective. I suggest submitting it to Communications Physics instead.

Reviewer #2 (Remarks to the Author):

In the revised manuscript, the authors have addressed all issues that I raised, and greatly increased the clarity of their manuscript. In my view it is now suitable for publication.

With regards to the novelty of the work, I believe that this is the first paper to transfer the idea of wide field of view into from optics into acoustic metamaterial lenses. Combined with the strong experimental demonstration, this makes a clear case for publication in Nature Communications.

David Powell
UNSW Canberra

Reviewer #3 (Remarks to the Author):

After a thorough examination of the revised manuscript and the authors' responses, it is acknowledged that significant efforts have been made to enhance the explanation of the underlying physics of the wide field of hearing (FOH). Despite these improvements, substantial contributions still could not be found to meet the standard of Nature Communications.

The authors claim that they emphasize the necessity of wide FOH along with experimental realization and add discussions on symmetry conversion, which had not been previously undertaken. Additionally, the authors provided guidelines to quantify the lens' performance.

Introducing a concept in optics into the acoustic community could be a contribution. However, without any modifications to make it applicable to other domain, the introduction itself cannot be considered to have a significant impact. As agreed by the authors, there were no substantial differences in the two fields of optics and acoustics concerning this wavefront engineering. In addition, the realization of the wide FOH lenses could be possible using alternative unit structures, e.g. Helmholtz resonator-based structures in [J. Phys. D, 52, 385303 (2019)], by simply employing the phase profile for wide FOH: the reviewer raised these concerns, yet the authors have not responded. The discussions on symmetry conversion, although helpful for conveying the concept of FOH and not having been undertaken previously, are not perceived as significantly impactful due to the lack of new findings on the underlying physics.

As for the establishment of quantitative criteria, the quantitative comparison among acoustic metalenses was not fully executed in this study. In particular, FOH capability—defined as the coverage of angles of incidence—for other lenses is marked as 'N/A', except for one in [S12]. Therefore, it is difficult to recognize the differences in FOH capability between the proposed lens and the existing others designed without considering the concept of FOH.

Response to Reviewers' Comments

(NCOMMS-23-37112A)

Title:

Wide field-of-hearing metalens for aberration-free sound capture

Authors:

Dongwoo Lee*, Beomseok Oh*, Jeonghoon Park, Seong-Won Moon, Kilsoo Shin, Sea-Moon Kim, and Junsuk Rho†

Overall remarks:

Dear Reviewers,

We greatly appreciate again the referees' comments throughout the review process. All the points raised by the reviewers have been incorporated into the revised manuscript. "Responses" and "modifications" made in this revision are colored in "blue" and "orange", respectively.

Response to Reviewer #1's comments:

Comment:

Although the authors have made great efforts to improve the quality of the manuscript, I still believe that it cannot meet the requirements of NC from the novelty perspective. I suggest submitting it to Communications Physics instead.

Reply:

We appreciate the reviewer's acknowledgment of our dedication to enhancing the quality of the manuscript throughout the first round of revision. While we are grateful for the suggestion to transfer the submission to another Nature portfolio journal, we would like to maintain our stance that our proposed work is still suitable for publication in NC. We respectfully contend that the reviewer's critique of our study is rooted in a vague claim that our work does not demonstrate sufficient novelty. Although the reviewer did not take our previous rebuttal into consideration, we have addressed the previous comments by providing additional support both in the main text and supplementary information (please refer to "Change"). We hope that the reviewer will consider our current rebuttal as well as our previous rebuttal.

Also, we place great emphasis on the novelty of our work again. The first demonstration/concept for wide FOH, which has not been reported elsewhere, is novel in itself. Moreover, the successful transfer of ideas across different wave domains has the potential to significantly expand our perspectives and deepen our understanding beyond each potential application. The evidence of some milestones, including but not limited to, is listed as follows:

1. Invisibility cloak

[Optics]

"Metamaterial electromagnetic cloak at microwave frequencies", Science 314, 977, 2006

"An optical cloak made of dielectrics", Nature Materials 8, 568, 2009

[Acoustics]

"Broadband acoustic cloak for ultrasound waves", Physical Review Letters 106, 024301, 2011

"Hiding under the carpet: a new strategy for cloaking", Physical Review Letters 101, 203901, 2008

2. Hyperlens

[Optics]

"Far-field optical hyperlens magnifying sub-diffraction-limited objects", Science 315, 1686, 2007

[Acoustics]

"Experimental demonstration of an acoustic magnifying hyperlens", Nature Materials 8, 931, 2009

3. Superlens

[Optics]

“Sub-diffraction-limited optical imaging with a silver superlens”, *Science* 308, 534, 2005

[Acoustics]

“Focusing ultrasound with an acoustic metamaterial network”, *Physical Review Letters* 102, 194301, 2009

“Negative refractive index and acoustic superlens from multiple scattering in single negative metamaterials”, *Nature* 525, 77, 2015

4. Dirac cone-assisted zero-refractive-index

[Optics]

“Dirac cones induced by accidental degeneracy in photonic crystals and zero-refractive-index materials”, *Nature Materials* 10, 582, 2011

[Acoustics]

“Observation of acoustic Dirac-like cone and double zero refractive index”, *Nature Communications* 8, 14871, 2017

5. Bianisotropy

[Optics]

“Theorems of bianisotropic media”, *Proceedings of the IEEE* 60, 1036, 1972

“High performance bianisotropic metasurfaces: asymmetric transmission of light”, *Physical Review Letters* 113, 023902, 2014

[Acoustics]

“Acoustic omni meta-atom for decoupled access to all octants of a wave parameter space”, *Nature Communications* 7, 13012, 2016

“Experimental evidence of Willis coupling in a one-dimensional effective material element”, *Nature Communications* 8, 15625, 2017

6. Bound states in the continuum

[Optics]

“Lasing action from photonic bound states in the continuum”, *Nature* 541, 196, 2017

[Acoustics]

“Sound trapping in an open resonator”, *Nature Communications* 12, 4819, 2021

We firmly believe that the exchange of insights across various wave domains is paramount. Our pioneering efforts to introduce a novel functionality, demonstrating wide FOH capabilities, have not been fully acknowledged, warranting greater recognition. Moreover, current research in the acoustic metalens community overlooks critical parameters such as MTF, NA, aberrations, and focusing efficiency. Addressing these considerations could significantly advance acoustic metalens technology.

Change:

1. Please refer to “Introduction” of the revised manuscript.

“We experimentally demonstrate the WFOH capability with highly sound-transparent and judicious phase-modulated metalens. By employing the perfect acoustic symmetry conversion, we realize an off-axis aberration-free compact sound sensing system with a reduced aperture-to-hearing distance and focal position area. Our findings have significant implications for various advanced applications, such as wide-angle acoustic sensing with high sensitivity, particle manipulation, as well as medical treatments involving high-intensity focused ultrasound (HIFU) and imaging. The successful implementation of WFOH opens up new avenues for further exploring these applications and achieving improved performance.”

2. Please refer to “Discussion” of the revised manuscript.

“In summary, we introduce a new metric, FOH—previously overlooked in existing acoustic metalenses—and showcase the hallmark of WFOH capability for the first time. Our proposed metalens is designed to capture and focus sound while reducing off-axis aberrations such as coma and field curvature. This is achieved through acoustic symmetry conversion, allowing for WFOH capability across an extensive angular range of approximately 140° without the need for the angle-dispersive configuration typically required in conventional metalenses. Furthermore, the use of a planar receiving plane with a short aperture-to-hearing plane distance facilitates compact wide-angle sound capture. Although this feature is realized for high-fidelity source-tracking in the audible range, it also offers potential extensions to its applicability in ultrasonic and submerged environments. The versatility and small form factor of an acoustic single-layer metalens, combined with its WFOH capability, make it well-suited for various scenarios where efficient sound capture, imaging capabilities, and wireless communication are required. The exploration of high-frequency regimes, particularly within the medical field, is of great interest. It involves addressing impedance mismatch and resolving the trade-off between the boundary layer effect and meta-atom period (see Supplementary Note 10 for details). Moreover, an improved FOH capability through the use of bianisotropy and non-locality with various meta-atom designs is highly expected (see Supplementary Note 11 for details). While our current experimental demonstration is focused on one-dimensional audible sound focusing, extending this into two and three dimensions holds promise for exploring novel applications. We believe that our work paves the way for further advancements in acoustic metalens technology and engineering, opening up new opportunities in the field of WFOH sound reception without complex and high-powered electronic components. We also note that the outlook of the WFOH towards wideband and

achromatic characteristics remains an open-ended question.”

3. Please refer to lines 211-213 on page 9 of the revised manuscript.

“..., Additionally, we calculate the peak signal-to-noise ratio (PSNR) to quantify the focusing performance⁶⁹⁻⁷¹, obtaining an average PSNR of 20.32 dB, indicating high-fidelity sound capturing performance (see details in Supplementary Note 9).”

4. Please refer to page S15 of the supplementary information (Supplementary Note 9, which is newly added).

“Supplementary Note 9: Peak signal-to-noise ratio

Peak signal-to-noise ratio (PSNR) stand as a widely-recognized metric for evaluating image quality [S10, S11], offering a quantitative measure of fidelity between an reference (or ground truth) signal and a real (or measured) signal. It is calculated as a function of the mean-squared error (MSE) between the reference and measured PSFs, providing a fundamental measure for quantitative assessment of focusing quality. PSNR can be expressed as follows [S11, S12]:

$$\text{PSNR} = 20 \log_{10}(\text{Max}_i/\sqrt{\text{MSE}}),$$

$$\text{MSE} = \frac{1}{n} \sum_{i=1}^n (\bar{I}_i - I_i)^2$$

where \bar{I}_i is the reference PSF, I_i is the measured PSF, and MAX_i is the maximum value of the measured PSF, and n is the total number of samples. As the MSE approaches zero, indicating minimal deviation from the reference, the PSNR becomes infinity. This signifies a well-defined PSF and implies high-quality wave focusing.

To evaluate the focusing quality of our system, we calculate the PSNR for incident angles ranging from 0 to 70 degrees (see Table S1). The calculated high PSNRs, even at large AOIs, provide clear evidence of the capability of our WFOH metalens that can be used for high-sensitivity wide-angle acoustic sensing.”

4. Please refer to page S16-S17 of the supplementary information (Supplementary Note 10, which is newly added).

“Supplementary Note 10: Impact of thermoviscous loss and meta-atom period on the focusing performance

Boundary-layer effects on transmission of meta-atoms

We examine the effects of thermoviscous loss and meta-atom period on the focusing performance. Thermoviscous effects become notable when the width of the acoustic channel is reduced to less than two orders of magnitude compared to the thickness of the viscous and thermal boundary layers [S13, S14]. The effects of viscous and thermal boundary layers at the solid-fluid interface result in losses. The thickness of viscous and thermal boundary layers is defined by $\delta_v = \sqrt{2\mu/\omega\rho_0}$, $\delta_t = \delta_v/\sqrt{\text{Pr}}$ where μ is the dynamic viscosity, ω is the angular frequency, ρ_0 denotes the mass density of the medium, and Pr is Prandtl number [S9, S13-S15]. At 3.43 kHz in air, δ_v is around 32 μm . In Fig. S9(a), we present the transmission of meta-atoms considering thermoviscous effects. The average transmission in the lossless case is 0.95, and 0.79 in the presence of thermoviscous loss. Although the loss is incorporated, we find that the majority of meta-atoms exhibit $T > 0.7$. Additionally, Fig. S9(b) shows the ratio between the viscous boundary layer and the channel width ($\Gamma = \delta_v/w$). For most meta-atoms, $\Gamma < 0.02$, suggesting that w is two orders of magnitude larger than the boundary layer thickness. As a result, the impact of thermoviscous loss is not significant enough to hinder efficient wave focusing. Indeed, the calculated PSNR values are sufficiently high (refer to Table S1), verifying that acoustic signals can be clearly resolved with high sensitivity.

The relationship between thermoviscous effects and meta-atom period on the focusing performance

In principle, there exists a trade-off between the spatial spacing (or period, denoted as $p = h_0$ in our work) of discretized meta-atoms and thermoviscous loss. As the period p decreases, the metalens may operate closer to a continuous phase, potentially resulting in smaller aberrations and higher focusing efficiency. However, this proximity to continuity introduces the influence of thermoviscous effects, which can diminish efficiency due to lower transmission. Conversely, increasing p alleviates thermoviscous loss; however, larger values of p induce parasitic diffraction, consequently diminishing focusing performance.

To explore the relationship between thermoviscous effects and meta-atom period on focusing performance, we conduct numerical calculations of the PSFs relative to p , considering the thermoviscous effects. In Fig. S10, we illustrate the normalized focusing intensity (without and with thermoviscous loss effects) and the average ratio between the viscous boundary layer and the channel width of meta-atoms (Γ_{avg}) of WFOH metalenses with periods ranging from $\lambda/9$ to $\lambda/14$. In the case without loss, we can clearly see that as p decreases, the metalens operates with a nearly-continuous phase, leading to a monotonic increase in intensity up to the theoretical limit.

However, upon considering thermoviscous loss, we observe a trade-off between thermoviscous effects, which are influenced by the channel width corresponding to the meta-atom period, and discrete effects. This trade-off suggests the existence of an optimal period. Such analysis is particularly important when designing acoustic metalenses in the ultrasonic range, where careful consideration of thermoviscous effects becomes imperative.”

Fig. S9. (a) Transmission of meta-atoms without loss (blue) and with thermoviscous loss (red). The horizontal dashed line indicates $T = 0.7$. (b) Ratio between the viscous boundary layer and the channel width of meta-atoms, defined by $\Gamma = \delta_v/w$.

Fig. S10. Effects of thermoviscous loss and meta-atom period on the metalens performance. Black lines with circle (square) markers represent the normalized focusing intensity without (with) thermoviscous effects, while the red line indicates the average ratio between the viscous boundary layer and the channel width of meta-atoms. For simplicity, the x-axis is represented as the inverse of the period.

Response to Reviewer #2's comments:

Comment:

In the revised manuscript, the authors have addressed all issues that I raised, and greatly increased the clarity of their manuscript. In my view it is now suitable for publication.

With regards to the novelty of the work, I believe that this is the first paper to transfer the idea of wide field of view into from optics into acoustic metamaterial lenses. Combined with the strong experimental demonstration, this makes a clear case for publication in Nature Communications.

David Powell
UNSW Canberra

Reply:

We appreciate the reviewer for the helpful evaluation and support of our work. As noted by the reviewer, our work is distinguished as the first paper to successfully transfer the concept of wide FOV into acoustic metalenses. We are confident that our work will be a significant milestone in introducing the functionality of FOH, thus establishing a solid foundation for further advancements and exploration in this field. Sincere thanks to the reviewer for the constructive comments throughout this process.

Response to Reviewer #3's comments:

Comment:

After a thorough examination of the revised manuscript and the authors' responses, it is acknowledged that significant efforts have been made to enhance the explanation of the underlying physics of the wide field of hearing (FOH). Despite these improvements, substantial contributions still could not be found to meet the standard of Nature Communications.

Reply:

We sincerely thank Reviewer #3 for revisiting and evaluating our work. We are grateful for the acknowledgment of the extensive efforts dedicated to enhancing the quality of the manuscript during the first round of revision. Although we understand and respect the reviewer's perspective that our work may not align with the standard of NC, we firmly believe that our proposed work is indeed suitable for publication in this esteemed journal. Below, we provide a more comprehensive elaboration on the novelty and importance of our findings.

Comment 1:

The authors claim that they emphasize the necessity of wide FOH along with experimental realization and add discussions on symmetry conversion, which had not been previously undertaken. Additionally, the authors provided guidelines to quantify the lens' performance. Introducing a concept in optics into the acoustic community could be a contribution. However, without any modifications to make it applicable to other domain, the introduction itself cannot be considered to have a significant impact. As agreed by the authors, there were no substantial differences in the two fields of optics and acoustics concerning this wavefront engineering.

Reply 1:

We sincerely thank the reviewer's feedback on our manuscript regarding the novelty of our research. We appreciate the opportunity to address your concerns and provide further clarification on the unique contributions of our study.

We are of the opinion that the significance of our work might be underestimated, particularly in light of the reviewer's perspective on the statement "*without any modifications to make it applicable to other domain, the introduction itself cannot be considered to have a significant impact*". One aspect we would like to emphasize is the innovative approach of transferring concepts from the field of optics to the domain of acoustics. While a wide FOV has been proposed in optical metalenses, its translation into the realm of acoustics itself represents a novel and unconventional approach. We believe that the first introduction of the WFOH metalenses, combined with a strong experimental demonstration represents a clear 'breakthrough' in the acoustic domain. The importance of this metric, which has never been seen before, is further emphasized by its prior absence in the evaluation of existing acoustic metalenses. The incorporation of the FOH capability as a new degree of freedom

and functionality will undoubtedly flourish in the design of acoustic metalenses going forward, following our successful demonstration and unveiling. It is not appropriate to discredit the novelty of an idea solely based on a lack of modification to existing physical principles. We have made a new chapter by introducing the FOH concept, which has not been observed before in the acoustic domain. The timely introduction of this new concept serves as our signature and proof of contribution. Thus, we respectfully suggest that the evaluation on the novelty of our work should focus on these aspects, rather than on the use of a specific phase profile.

In summary, from these viewpoints, we would like to highlight the scientific impact and significance of our ‘field-opening work’, and we anticipate that such contributions will lead to emergence of more advancements. Additionally, we provide the following milestones of successful idea exchange/transfer, currently in the spotlight between two different domains:

1. Invisibility cloak

[Optics]

“Metamaterial electromagnetic cloak at microwave frequencies”, *Science* 314, 977, 2006

“An optical cloak made of dielectrics”, *Nature Materials* 8, 568, 2009

[Acoustics]

“Broadband acoustic cloak for ultrasound waves”, *Physical Review Letters* 106, 024301, 2011

“Hiding under the carpet: a new strategy for cloaking”, *Physical Review Letters* 101, 203901, 2008

2. Hyperlens

[Optics]

“Far-field optical hyperlens magnifying sub-diffraction-limited objects”, *Science* 315, 1686, 2007

[Acoustics]

“Experimental demonstration of an acoustic magnifying hyperlens”, *Nature Materials* 8, 931, 2009

3. Superlens

[Optics]

“Sub-diffraction-limited optical imaging with a silver superlens”, *Science* 308, 534, 2005

[Acoustics]

“Focusing ultrasound with an acoustic metamaterial network”, *Physical Review Letters* 102, 194301, 2009

“Negative refractive index and acoustic superlens from multiple scattering in single negative metamaterials”, *Nature* 525, 77, 2015

4. Dirac cone-assisted zero-refractive-index

[Optics]

“Dirac cones induced by accidental degeneracy in photonic crystals and zero-refractive-index materials”, *Nature Materials* 10, 582, 2011

[Acoustics]

“Observation of acoustic Dirac-like cone and double zero refractive index”, Nature Communications 8, 14871, 2017

5. Bianisotropy

[Optics]

“Theorems of bianisotropic media”, Proceedings of the IEEE 60, 1036, 1972

“High performance bianisotropic metasurfaces: asymmetric transmission of light”, Physical Review Letters 113, 023902, 2014

[Acoustics]

“Acoustic omni meta-atom for decoupled access to all octants of a wave parameter space”, Nature Communications 7, 13012, 2016

“Experimental evidence of Willis coupling in a one-dimensional effective material element”, Nature Communications 8, 15625, 2017

6. Bound states in the continuum

[Optics]

“Lasing action from photonic bound states in the continuum”, Nature 541, 196, 2017

[Acoustics]

“Sound trapping in an open resonator”, Nature Communications 12, 4819, 2021

Comment 2:

In addition, the realization of the wide FOH lenses could be possible using alternative unit structures, e.g. Helmholtz resonator-based structures in [J. Phys. D, 52, 385303 (2019)], by simply employing the phase profile for wide FOH: the reviewer raised these concerns, yet the authors have not responded. The discussions on symmetry conversion, although helpful for conveying the concept of FOH and not having been undertaken previously, are not perceived as significantly impactful due to the lack of new findings on the underlying physics.

Reply 2:

We apologize for not providing a clear response in the previous rebuttal. Respectfully, it is worth noting that a considerable amount of time has passed since the initial proposal of the first optical wide field-of-view metalens in 2017, without any corresponding research exploration in the acoustic domain. The progress in exchanging and transferring ideas between the optical and acoustic metalens communities has been hindered by the limited understanding of imaging and sensing capabilities in the acoustic domain, to our knowledge. While acoustic focusing metalenses for proof-of-concept purposes have been conducted thus far, there has been a lack of in-depth understanding regarding the imaging and sensing potential of acoustic metalenses. For this reason, the unique functionality of the FOH has not been reported, despite the growing importance of the FOV merit and the increased attention on wide FOV metalenses, driven by the demands such as single-chip wide-angle imaging,

3D sensing, AR/VR devices. [R1, R2]. None of the existing works on the acoustic side have considered important factors for imaging/sensing capabilities, including MTF, NA, aberrations, and focusing efficiency. In addition, we have provided a detailed clarification of the acoustic symmetry conversion mechanism, which has been articulated more clearly than in wide FOV metalens in optics, as agreed by the reviewer.

For the reviewer's statement "*the realization of the wide FOH lenses could be possible using alternative unit structures, e.g. Helmholtz resonator-based structures in [J. Phys. D, 52, 385303 (2019)]*", we agree with your perspective. The design of meta-atoms for acoustic metalenses is not limited to a specific single solution, and there are numerous possibilities to explore. It is essential to engage in extensive research and consider diverse design strategies to advance the development of WFOH metalenses. Beyond the design perspective which have further elaborated in "Discussion" section of the revised manuscript (please refer to "Change 2"), we newly introduced a FOH metric and experimentally demonstrated unprecedented WFOH capability successfully.

[R1] Luo, XianGang, et al. "Recent advances of wide-angle metalenses: principle, design, and applications." *Nanophotonics* 11.1 (2021): 1-20.

[R2] Yang, Fan, et al. "Wide field-of-view metalens: a tutorial." *Advanced Photonics* 5.3 (2023): 033001-033001.

Change 2:

1. Please refer to "Introduction" of the revised manuscript.

"We experimentally demonstrate the WFOH capability with highly sound-transparent and judicious phase-modulated metalens. By employing the perfect acoustic symmetry conversion, we realize an off-axis aberration-free compact sound sensing system with a reduced aperture-to-hearing distance and focal position area. Our findings have significant implications for various advanced applications, such as wide-angle acoustic sensing with high sensitivity, particle manipulation, as well as medical treatments involving high-intensity focused ultrasound (HIFU) and imaging. The successful implementation of WFOH opens up new avenues for further exploring these applications and achieving improved performance."

2. Please refer to "Discussion" of the revised manuscript.

"In summary, we introduce a new metric, FOH—previously overlooked in existing acoustic metalenses—and showcase the hallmark of WFOH capability for the first time. Our proposed metalens is designed to capture and focus sound while reducing off-axis aberrations such as coma and field curvature. This is achieved through acoustic symmetry conversion, allowing for WFOH

capability across an extensive angular range of approximately 140° without the need for the angle-dispersive configuration typically required in conventional metalenses. Furthermore, the use of a planar receiving plane with a short aperture-to-hearing plane distance facilitates compact wide-angle sound capture. Although this feature is realized for high-fidelity source-tracking in the audible range, it also offers potential extensions to its applicability in ultrasonic and submerged environments. The versatility and small form factor of an acoustic single-layer metalens, combined with its WFOH capability, make it well-suited for various scenarios where efficient sound capture, imaging capabilities, and wireless communication are required. The exploration of high-frequency regimes, particularly within the medical field, is of great interest. It involves addressing impedance mismatch and resolving the trade-off between the boundary layer effect and meta-atom period (see Supplementary Note 10 for details). Moreover, an improved FOH capability through the use of bianisotropy and non-locality with various meta-atom designs is highly expected (see Supplementary Note 11 for details). While our current experimental demonstration is focused on one-dimensional audible sound focusing, extending this into two and three dimensions holds promise for exploring novel applications. We believe that our work paves the way for further advancements in acoustic metalens technology and engineering, opening up new opportunities in the field of WFOH sound reception without complex and high-powered electronic components. We also note that the outlook of the WFOH towards wideband and achromatic characteristics remains an open-ended question.”

Comment 3:

As for the establishment of quantitative criteria, the quantitative comparison among acoustic metalenses was not fully executed in this study. In particular, FOH capability—defined as the coverage of angles of incidence—for other lenses is marked as 'N/A', except for one in [S12]. Therefore, it is difficult to recognize the differences in FOH capability between the proposed lens and the existing others designed without considering the concept of FOH.

Reply 3:

We appreciate the reviewer’s comment. The use of ‘N/A’ notation in Table S2 was intended to indicate that previous studies did not specifically analyze the effect of oblique incidence as well as the critical metrics to characterize the performance of the lens. Therefore, the table was constructed directly based on the content provided in references, which resulted in the inclusion of ‘N/A’ where relevant data was not available or not addressed in those studies. We entirely agree the statement *‘it is difficult to recognize the differences in FOH capability between the proposed lens and the existing others designed without considering the concept of FOH’*. In response to this, a quantitative analysis is conducted to compare the FOH capability of conventional and

WFOH metalenses, specifically in relation to numerical aperture. The obtained results have been included in Supplementary Note 11. Additionally, we have made revisions to Table S2 in order to provide a more coherent comparison and improve the understanding of the significance of our work, as well as its differentiation from previous studies.

Change 3:

1. Please refer to page S18-S19 of the supplementary information (Supplementary Note 11).

“Supplementary Note 11: Field-of-hearing characterization and comparison with previous studies on acoustic metalenses

We compare our work with previous studies on acoustic focusing using planar metasurfaces. A detailed comparison, as shown in Table S2, highlights crucial performance metrics of the lens, particularly for the FOH capability. To our knowledge, the determination of the size and focal length of metalenses seems arbitrary in Refs [S16-S31], indicating a lack of consideration for NA during the design process. This deficiency is often a result of the predominant focus on proof-of-concept demonstrations of focusing capabilities through field measurements. In contrast to the existing limitations of acoustic metalenses that encounter difficulties in achieving WFOH capabilities using the hyperbolic phase profile (commonly recognized as the trade-off between high-NA and wide FOV in conventional metalenses [S32]), our current work significantly enhances wide-angle focusing capability—an aspect often overlooked in previous studies—by emphasizing the necessity of considering the FOH. While Ref. [S18] it lacked detailed interpretation with its narrow FOH.

Directly comparing the performance between previous studies and our work is laborious, as existing metalenses have often overlooked the FOH capability related to imaging evaluation metrics. Therefore, we calculate the ideal FOH as a function of the NA for both hyperbolic and quadratic phase-based metalenses. To quantitatively characterize the FOH, we introduce a new function that evaluates the differences between the PSFs obtained under normal and oblique incidence [S33, S34], which can be defined by

$$e(\theta_i) = \int |I(y, 0)/\sqrt{\eta_0} - I(y - \Delta y, \theta_i)/\sqrt{\eta_i}|^2 dy, \quad (\text{S44})$$

where $I(y, 0)$ and $I(y - \Delta y, \theta_i)$ indicate the PSF at normal and oblique incidence, respectively. Δy denotes the spacing required to align the peak center with the reference $I(y, 0)$, and η indicates the focusing efficiency which is defined by $\eta = \left[\int_{-y}^y I|_{x=f} dy \right] / \left[\int_{-D/2}^{D/2} I_0|_{x=0} dy \right]$ integrated over the $[-y, y] = 3 \times \text{FWHM}$ region at the focal spot for each different AOIs (refer to the main text), where I_0 is the incident intensity and I is the focused intensity. We define the

FOH as twice the AOI when $e(\theta_i)$ is smaller than 0.1, indicating the formation of a high-quality focusing spot.

To illustrate the calculation process of $e(\theta_i)$ schematically, we depicted an example of PSFs of a conventional metalens with an NA of 0.5 and arbitrary AOIs (Fig. S11(a)). The left and right panels indicate the PSFs and the aligned PSFs, respectively. In Fig. S11(b), we present the FOH performance of both conventional and WFOH metalenses, similar to findings in Refs. [S33, S34]. In particular, we observe that the FOH of the conventional metalens decreases as the NA increases, while the FOH of the WFOH metalens exhibits an extension towards the theoretical limit of 180° . In this study, we design our metalenses based on an NA of 0.9 and achieve the FOH of 140° . The achieved FOH can be mainly attributed to the limited angular responses of the proposed meta-atoms (see Fig. S5 in Supplementary Note 5). To achieve WFOH performance closer to the ideal case, it is anticipated that considering bianisotropic and non-local effects [S4, S35] would be beneficial. However, it remains challenging to obtain such performance using conventional metalenses.

Existing metalenses [S16-S31] have not accounted for or examined oblique incidence, making direct comparisons of FOH performance with our work difficult. However, based on the analysis presented in Fig. S11, it is evident that conventional phase-based single-layer metalenses are unlikely to achieve the WFOH capability.”

Fig. S11. (a) Schematic illustration of the PSFs and the aligned PSFs with different AOIs. (b) Achievable FOH with respect to NA for WFOH (red) and conventional (blue) metalenses.

Table S2. Summary of acoustic metalenses.

Ref.	Lens type	Numerical aperture	Consideration of oblique incidence	FOH (Exp./Num.)	Meta-atom thickness (λ)	Full-width at half maximum (λ)	Focusing efficiency (%)	Operating frequency (kHz)
This work	2D/ Transmissive	0.9	O	140 / 140	0.75	0.39-0.42	41-43	3.43
[S16]	2D/ Transmissive	0.76	X	-	0.5	0.56	-	2.5, 4, 5.5
[S17]	2D/ Transmissive	0.74	X	-	0.54	0.52	41.5	3.1
[S18]	2D/ Reflective	0.51	O	- / 20	0.5	-	-	2.8-5.6
[S19]	2D/ Transmissive	0.49	X	-	0.87	-	-	7.5
[S20]	3D/ Transmissive	0.5	X	-	0.05	0.98	-	3.43
[S21]	2D/ Transmissive	0.68	X	-	0.5	-	-	2.7, 4.1, 5.5
[S22]	2D/ Transmissive	0.8, 0.94	X	-	0.65	0.52, 0.39	-	5
[S23]	3D/ Reflective	0.68, 0.73	X	-	0.07	0.76, 0.74	-	3.43
[S24]	2D/ Reflective	0.7	X	-	0.2	0.64-0.75	-	3, 3.5, 4.5
[S25]	2D/ Reflective	0.15	X	-	0.25	0.62	-	500 (water)
[S26]	3D/ Transmissive	0.87	X	-	0.98	0.55-0.83	-	8
[S27]	2D/ Reflective	0.86	X	-	0.5	-	-	1.7, 3.4, 5.1
[S28]	2D/ Reflective	0.78	X	-	0.21	-	-	2
[S29]	2D/ Transmissive	0.79	X	-	0.87	0.59-0.7	-	1-4
[S30]	2D/ Reflective	0.68	X	-	0.25	-	-	1.3, 1.9, 2.4
[S31]	3D/ Transmissive	0.88	X	-	0.8	~0.6	-	5.5